



# Automated detection and monitoring of methane super-emitters using satellite data

Berend J. Schuit[1,2], Joannes D. Maasakkers[1], Pieter Bijl[1], Gourav Mahapatra[1], Anne-Wil van den Berg[1,†], Sudhanshu Pandey[1,‡], Alba Lorente[1], Tobias Borsdorff[1], Sander Houweling[1,3], Daniel J. Varon[2,4], Jason McKeever[2], Dylan Jervis[2], Marianne Girard[2], Itziar Irakulis-Loitxate[5,9], Javier Gorroño[5], Luis Guanter[5,6], Daniel H. Cusworth[7,8], and Ilse Aben[1,3]

[1]SRON Netherlands Institute for Space Research, Leiden, The Netherlands
[2]GHGSat Inc, Montréal, Canada
[3]Department of Earth Sciences, Vrije Universiteit Amsterdam, Amsterdam, The Netherlands
[4]Harvard University, Cambridge, MA, US
[5]Research Institute of Water and Environmental Engineering (IIAMA), Universitat Politècnica de València (UPV), Valencia, Spain
[6]Environmental Defense Fund, Amsterdam, The Netherlands
[7]Carbon Mapper, Pasadena, CA, US
[8]University of Arizona, Tucson, AZ, US
[9]International Methane Emission Observatory, United Nations Environment Program, Paris, France
[†]Now at: Department of Meteorology and Air Quality, Wageningen University, Wageningen, The Netherlands
[‡]Now at: Jet Propulsion Laboratory, California Institute of Technology, Pasadena, CA, US

**Correspondence:** Berend J. Schuit (B.J.Schuit@sron.nl)

**Abstract.**

A reduction in anthropogenic methane emissions is vital to limit near-term global warming. A small number of so-called super-emitters is responsible for a disproportionally large fraction of total methane emissions. Since late 2017, the TROPOspheric Monitoring Instrument (TROPOMI) has been in orbit providing daily global coverage of methane mixing ratios at a resolution

of up to 7x5.5 km$^2$, enabling the detection of these super-emitters. However, TROPOMI produces millions of observations each day, which together with the complexity of the methane data, makes manual inspection infeasible. We have therefore designed a two-step machine learning approach using a Convolutional Neural Network to detect plume-like structures in the methane data and subsequently apply a Support Vector Classifier to distinguish emission plumes from retrieval artefacts. The models are trained on pre-2021 data, and subsequently applied to all 2021 observations. We detect 2974 plumes in 2021 with a mean

estimated source rate of 44 t h$^{-1}$ and 5-95$^{th}$ percentile range of 8-122 t h$^{-1}$. These emissions originate from 94 persistent emission clusters and hundreds of transient sources. Based on bottom-up emission inventories, we find that most detected plumes are related to urban areas / landfills (35%), followed by plumes from gas infrastructure (24%), oil infrastructure (21%) and coal mines (20%). For twelve (clusters of) TROPOMI detections, we "tip-and-cue" targeted observations and analysis of high-resolution satellite instruments to identify the exact sources responsible for these plumes. Using high-resolution observations

from GHGSat, PRISM and Sentinel-2, we detect and analyze both persistent and transient facility-level emissions underlying the TROPOMI detections. We find emissions from landfills and fossil fuel exploitation facilities, for the latter we find up to ten



facilities contributing to one TROPOMI detection. Our automated TROPOMI-based monitoring system in combination with high-resolution satellite data allows for the detection, precise identification and monitoring of these methane super-emitters, which is essential for mitigating their emissions.



## 1 Introduction

Anthropogenic methane emissions have caused at least 25% of human-induced global warming (Ocko et al., 2018; IPCC, 2021). Methane's atmospheric concentration has increased by a factor of 2.5 since the pre-industrialized era (Szopa et al., 2021) and the rate of increase has accelerated in recent years (NOAA, 2022). Due to its relatively short atmospheric lifetime and large global warming potential (81 times that of $CO_2$ over a timespan of 20 years (IPCC, 2021)), methane has an important role in the rate of climate warming (Nisbet et al., 2020; Ocko et al., 2021; Szopa et al., 2021). Reducing global methane emissions is therefore vital to achieve the goals set out in the 2015 Paris climate agreement (Nisbet et al., 2020). Since November 2021, over 125 countries signed the Global Methane Pledge (European Commission, 2021; CCAC, 2022), committing them to reduce their methane emissions by 30% in 2030 compared to 2020 levels, this could avoid 0.2°C of global mean warming by 2050 (CCAC, 2022; UNEP and CCAC, 2021). In order to reduce global methane emissions fast and effectively during this decade, it is paramount to identify the largest anthropogenic sources of methane and mitigate those. We therefore propose an automated detection and monitoring system using satellite data with machine learning models to detect methane super-emitters.

The dominant anthropogenic methane emission sources are agriculture (livestock and rice cultivation), oil and gas exploitation, waste management and coal mining; the exact locations and magnitudes of emissions are still uncertain (Saunois et al., 2020). Large fractions of methane emissions in various sectors could be mitigated using existing technology, about a quarter of those at no net cost (Nisbet et al., 2020; Ocko et al., 2021; Lauvaux et al., 2022). Moreover, a small number of emitters is responsible for a disproportionally large fraction of total anthropogenic emissions (Zavala-Araiza et al., 2015; Jacob et al., 2016). These concentrated point sources are often referred to as 'super-emitters' and are difficult to account for in global bottom-up inventories (Zavala-Araiza et al., 2015) as they are often caused by severe malfunctioning or abnormal operating conditions, e.g. dysfunctional natural gas flaring systems (Irakulis-Loitxate et al., 2022a, b; Plant et al., 2022). Super-emitters are not limited to oil and gas production but also occur in the coal mining and waste sectors (Cusworth et al., 2020; Sadavarte et al., 2021; Maasakkers et al., 2022b). Detection, localization, and global monitoring of these methane super-emitters provides a large opportunity to reduce emissions (UNEP and CCAC, 2021; Parry et al., 2022).

One way to obtain more insight in where super-emitters occur is to perform measurements on the ground, with drones, or aircraft campaigns. Several regions with known frequent and large methane emissions have been mapped in detail with aircraft campaigns (e.g. Frankenberg et al., 2016; Duren et al., 2019; Yu et al., 2022; Plant et al., 2022). While ground-based or airborne measurements are limited in spatial and temporal coverage, satellite observations have the potential for global monitoring of methane point-sources with frequent revisits (Jacob et al., 2016; Cusworth et al., 2019; Jacob et al., 2022). The TROPOspheric Monitoring Instrument (TROPOMI) (Veefkind et al., 2012) was launched in 2017 and observes atmospheric methane column-averaged mixing ratios with a pixel size down to 7x5.5 km$^2$ and daily global coverage (Hu et al., 2018; Lorente et al., 2021), resulting in a point-source detection limit down to ∼5 t h$^{-1}$ under favorable conditions (Jacob et al., 2016). TROPOMI data have been used to quantify global (Qu et al., 2021) and country-level (Chen et al., 2022) distributions of methane emissions as well



as large area sources, such as oil and gas basins like the Permian Basin (Zhang et al., 2020; de Gouw et al., 2020; Schneising
et al., 2020; Shen et al., 2022). Lauvaux et al. (2022) performed a study into oil and gas related methane super-emitters using
TROPOMI data. Several individual super-emitters have been studied in detail using TROPOMI CH$_4$ data; e.g. natural gas well
blowouts (Pandey et al., 2019; Cusworth et al., 2021; Maasakkers et al., 2022a), and various persistent sources (Varon et al.,
2019; Sadavarte et al., 2021; Tu et al., 2022a, b).

Given the intermediate spatial resolution of TROPOMI, it can only be used to pinpoint the sources of emissions from the
largest and most isolated point sources. For more challenging sources, high-resolution instruments are more suitable to detect
and identify the exact location of super-emitters. So far, the only in-orbit satellite instrument specifically designed to do so are
the GHGSat instruments that have a spatial resolution of $\sim$25x25 m$^2$ over targeted $\sim$15x10 km$^2$ scenes (Varon et al., 2019;
Jervis et al., 2021; Ramier et al., 2020; MacLean et al., 2021). More recently it was shown that several Earth surface imagers
with spectral sensitivity in the short-wave infrared (SWIR), although not designed for this purpose, can detect signals from
methane super-emitters under favorable conditions. As such the hyperspectral PRISMA instrument (Cogliati et al., 2021) was
used to retrieve atmospheric methane plumes related to fossil fuel exploitation using targeted scenes of 30x30 km$^2$ at a spatial
resolution of 30 m (Guanter et al., 2021; Cusworth et al., 2021). Varon et al. (2021) demonstrated that the Multi-Spectral Instru-
ment, a band imaging instrument onboard the Sentinel-2 satellite (Drusch et al., 2012), is capable of retrieving large methane
plumes with a pixel resolution of 20 m for continuous, 290 km wide swaths over favorable terrain. TROPOMI's daily global
coverage is particularly well suited to guide observations of these high-resolution instruments, that often have limited viewing
domains, to identify large sources of methane at facility level (Irakulis-Loitxate et al., 2022b; Maasakkers et al., 2022b).

TROPOMI has collected over 5 years of methane data, which include numerous methane emission plume signals that cannot
feasibly be identified manually. To monitor the growing volume of data for super-emitters, an automated approach is needed.
The vast amount of data provides an opportunity for machine learning techniques that require a substantial amount of represen-
tative training data. Applications of machine learning in satellite remote sensing have mostly focused on studying the Earth's
surface, but also include monitoring anomalous atmospheric conditions and identify plume signatures in large datasets (Valade
et al., 2019; Finch et al., 2022). Detecting methane plumes in TROPOMI data is particularly challenging because not every
retrieved methane enhancement is a genuine methane emission plume as retrieval artefacts and natural variability can appear
like methane plumes. We therefore use a two-step machine learning method to identify methane emission plumes. We first use
a Convolutional Neural Network (CNN) to detect plume-like structures in TROPOMI XCH$_4$ data and then use a Support Vector
Classifier (SVC) to evaluate these potential plumes using additional information to distinguish real plumes from artefacts. We
train the machine learning models on verified TROPOMI methane plumes from 2018-2020, and then apply the trained models
to 2021 data. Based on the 2021 detections, we target three high-resolution satellite instruments (GHGSat-Cx, PRISMA and
Sentinel-2) to determine the origin of the emissions down to facility level. The combination of the automated global monitoring
based on TROPOMI with the high resolution of targeted instruments allows the detection and characterization of super-emitters
around the world.





## 2 Data & Methods

We use two machine learning models in sequence to detect plumes in the TROPOMI methane data. First we apply a Convolutional Neural Network to detect plume-like structures in TROPOMI methane atmospheric mixing ratio data, then we use additional atmospheric parameters and supporting data to further distinguish between genuine methane plumes and retrieval artefacts. We then use (targeted) high-resolution methane observations to pinpoint the responsible sources.

### 2.1 TROPOMI

We use data over land from the TROPOMI $XCH_4$ Level 2 scientific data product version 18_17. This product version is consistent with operational version 02.03.01 (Lorente et al., 2021; Hasekamp et al., 2022), but re-processed for the full timespan of the mission resulting in a homogeneous data product (SRON CH4 L2 team, 2022). We use albedo-bias corrected data with: QA≥0.4, methane precision < 10 ppb, SWIR aerosol optical depth < 0.13, NIR aerosol optical depth < 0.30, SWIR surface albedo > 0.02, mixed albedo (2.4 * NIR surface albedo - 1.13 * SWIR surface albedo) < 0.95, and SWIR cloud fraction < 100 0.02. The loosened filtering compared to the recommended QA=1 filter provides more coverage but also retains more biased retrievals, especially at the borders of clouds or along coasts. The methane data is de-striped following the approach introduced by Borsdorff et al. (2018).

To train a machine learning model to recognise methane plumes in TROPOMI data, we created a dataset of scenes consisting 105 of 32x32 pixels, both with and without methane plumes. 32x32 pixels correspond roughly to an area of ∼176x232 km$^2$ at nadir, up to ∼176x448 km$^2$ for larger viewing angles. For scenes with plumes, we use data over 60 persistently emitting locations identified using long-term wind-rotated averages (Maasakkers et al., 2022b). By manual inspection, we compile a dataset of 828 positive scenes from 2018-2020 with plumes, of which 195 originate from coal mines, 203 from landfills/urban areas, and 430 from oil&gas infrastructure. An example scene is shown in Figure 1a, including the local windfield at the time of 110 observation.





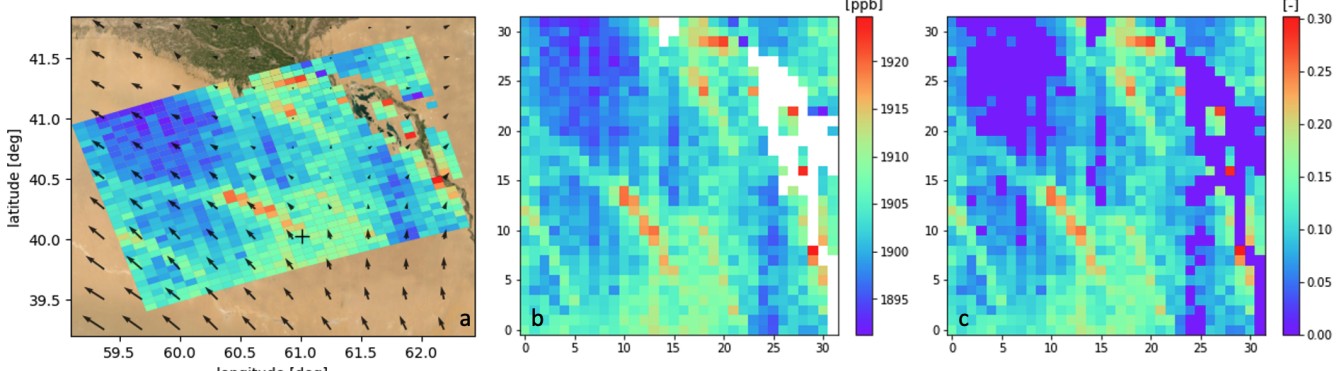

**Figure 1.** Atmospheric methane mixing ratios of a 32x32 pixel scene containing a methane plume originating from a known persistent source (indicated by the +) as observed by TROPOMI on 2021-12-05 at 08:47 UTC (not included in the training data). (a) Mercator projection of the scene over ESRI World Imagery (Esri, Maxar, Earthstar Geographics, and the GIS User Community, 2022), arrows show the local GEOS-FP 10m windfield (Molod et al., 2012). (b) 32x32 pixel scene in along-orbit vs across-orbit direction, indicating filtered pixels. (c) the same scene after pre-processing as used by the CNN.

A set of (negative) scenes without an emission signal was obtained through manual inspection of six full orbits in different sections of the orbital repeat cycle, covering a diverse set of surfaces and (meteorological) conditions. We obtain 32x32 pixel scenes using a moving window algorithm with 50% overlap, resulting in a dataset of 2242 scenes without a plume signal. The moving window algorithm later ensures that if a plume is cut in half in a particular scene, it will be in the center of the
adjacent -and partially overlapping- scene. Scenes with <20% valid XCH$_4$ pixels are discarded. This processing is applied to full orbits, the scenes with plumes originating from known locations were processed to match the same format. Combined, the dataset contains 3070 scenes used for training, the difference in the number of positive (828) and negative (2242) scenes is corrected for later on using class weights. For each scene, we store 46 other channels of supporting information from the same TROPOMI Level 2 methane dataproduct, including co-retrieved atmospheric properties, meteorological parameters and
geometric properties. These channels are used in later steps of the machine learning pipeline.

In order to correct for differences in local background concentrations (e.g. due to difference in latitude or surface altitude), each scene is normalized from 0 to 1. Values below the mean methane concentration of the scene minus one standard deviation are set to 0. Values above the mean plus 100 ppb minus one standard deviation are set to 1, values in between are linearly
distributed. Filtered pixels are set to 0. This pre-processing preserves the information of plume-like enhancements above the local background. Examples of the normalization input and output are shown in respectively Figure 1b and 1c.





## 2.2 Convolutional Neural Network (CNN)

We use a Convolutional Neural Network (LeCun et al., 2010) to detect methane plumes in the TROPOMI methane data, split into 32x32 pixel scenes. The training of a CNN can be done with image level labels (plume or no plume). There is no

need to indicate where within the image the feature of interest is located, the CNN learns to localize these features during training. The main advantages of the CNN compared to regular neural networks or other machine learning models are that: the CNN is capable of retaining spatial information which is lost in fully-connected layers or machine learning models like decision trees or support vector machines (Selvaraju et al., 2020); the same convolutional kernel scans the entire image, which is computationally efficient; and that the model is rotational and translational invariant when properly trained (LeCun et al.,

2010). This last model property is essential for the automated detection of plumes as those can be located anywhere within a scene and the wind can be in any direction. The CNN's output is a prediction between 0 and 1 indicating the confidence of the model about the presence of a plume-like structure. Scenes with prediction scores >0.5 are classified as plumes. As high-level architecture we selected two convolutional blocks followed by two fully-connected layers and an output node (Figure 2). We found that deeper networks (e.g. Resnet (He et al., 2016) or VGG-16 (Simonyan and Zisserman, 2014)) did not yield an

improvement in performance for this problem with relatively low resolution, small size (32x32 pixel) scenes.

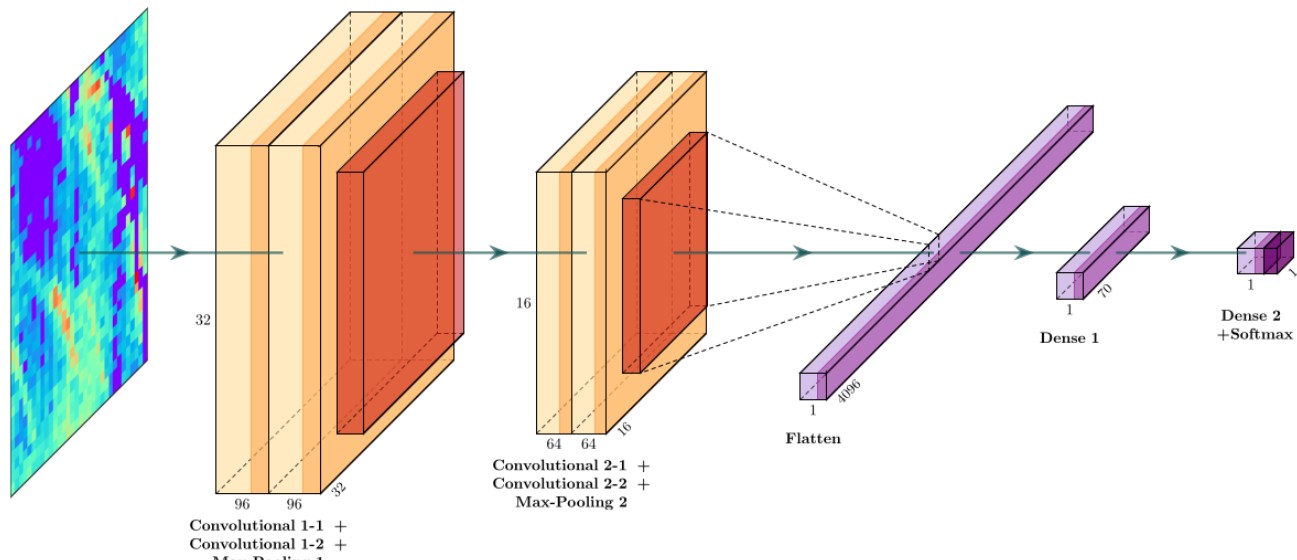

**Figure 2.** A schematic overview of the Convolutional Neural Network with a pre-processed 32x32 pixel TROPOMI methane scene (left) as input (Figure 1c). The CNN consists of two convolutional blocks (each with two Convolutional layers followed by a Max-Pooling layer) followed by two Dense (or Fully-Connected) layers and an output node. Numerical values show input dimensions, layer dimensions and optimized hyperparameter values. Visualization generated using PlotNeuralNet (Iqbal, 2018).





For the training process, the class weight parameter is set to the ratio between the number of plumes (828 positives) and negatives (2242), such that both categories have equal impact. These scenes mostly contain clear positives and clear negatives to effectively learn distinguishing features. We randomly split the data into a training (80%) and test (20%) set. 20% of the training scenes are then used as validation subset (Table A1). The validation dataset is used to infer whether there is a general-

ized performance increase during the training of the CNN to prevent overfitting. We then augment the data in order to obtain larger training and test sets; all scenes are rotated 90° thrice and flipped, enlarging the datasets by a factor 8.

We use the training dataset of 19,648 (augmented) scenes (Table A1) to train the CNN. The CNN was designed and trained using Keras (Chollet et al., 2015), at first we selected the default values for the hyperparameters, which control the details of

the high-level architecture and training process (i.e. number of layers, dimensions of layers, number of filters, learning rate etc.). The model is trained for a maximum of 100 epochs, optimizing the validation loss, using binary cross-entropy as the loss function and ADAM as the optimizer. We use a 0.4 dropout layer in the first fully-connected layer during training to make the model more robust. We apply the ReLU (rectified linear unit) the activation function in all layers except for the final layer where we apply softmax as the activation function. To force the model to focus on plume-like signatures, the loss

weight of plume scenes is set to double that of negatives scenes. Training is halted after the validation loss does not improve for several epochs and the best model weights are restored. After training, the model performance is inferred by classifying the labeled testset. After training this initial "default" model version, the hyperparameters were further optimized using Keras Tuner (O'Malley et al., 2019) and Hyperband (Li et al., 2018). The optimal hyperparameters depend on the size of the training dataset, architecture of the CNN, number of classes and problem type. The search space for the optimization was defined using

insights from the initial training, theoretical foundation, and design constraints. We inspected the hyperparameters of the top 10 performing models and selected the optimal hyperparameters by combining this optimization with expert judgement on this particular problem. Figure 2 shows a schematic overview of the CNN with optimized hyperparameters.

We evaluate the performance of the CNN using performance evaluation metrics (Equation 1) calculated from the number of

true positives and negatives (TP, TN), and false positives and negatives (FP, FN). The Cohen's Kappa score (Cohen, 1960) is a weighted accuracy which takes into account class imbalances and chance agreement. The Recall indicates which fraction of plumes present in the testset are correctly identified and the precision indicates which fraction of scenes identified as plumes are actually a plume, the F1-score incorporates both into a single metric.

$$\text{Accuracy} = \frac{TP+TN}{TP+TN+FP+FN} \qquad \text{Precision} = \frac{TP}{TP+FP} \qquad \text{Recall} = \frac{TP}{TP+FN} \qquad F_1 = 2 \cdot \frac{\text{precision} \times \text{recall}}{\text{precision} + \text{recall}}$$
(1)

To test the influence of the split of the training and test datasets (which can be an issue for datasets of limited size), the training of the model with optimized hyper-parameters was repeated 50 times with different splits. We found that model performance is relatively insensitive to different training splits with kappa = 0.943 ± 0.012, recall = 0.956 ± 0.014 and F1 = 0.958 ± 0.009





(standard deviations). We further found that small changes in the hyperparameters have even lower effect compared to different training splits. The consistent performance on the corresponding test datasets shows the model is robust and well generalized.

We focus on recall over precision, because the key focus is to have as few potential plumes as possible go undetected. We selected the model which scored best on kappa, second best on F1-score and third best on recall. The performance metrics of the selected CNN are shown in Figure 3a. Manual inspection of the misclassified scenes (30 false negatives and 37 false positives, out of 4912 augmented test scenes) indicates these are borderline cases with difficult to discern morphological structures that are even challenging to a human expert.

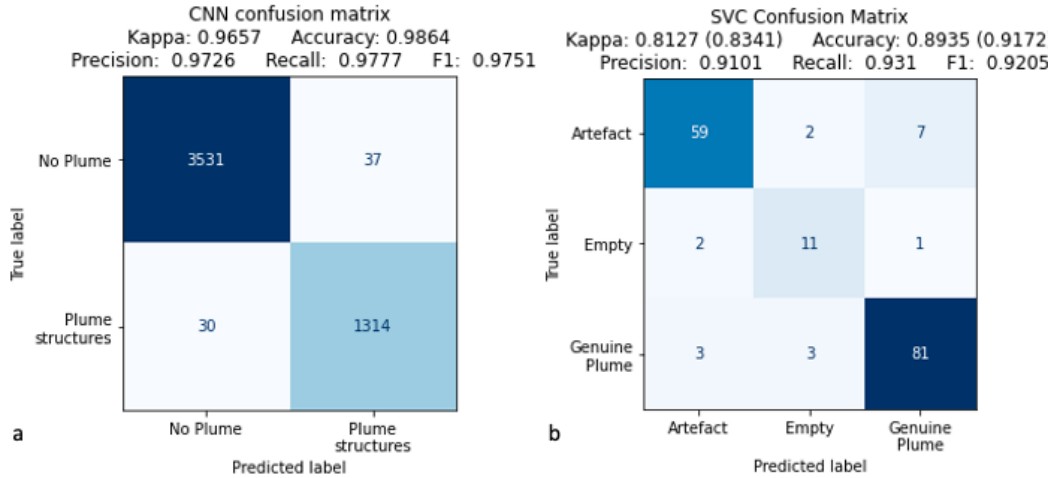

**Figure 3.** Confusion Matrices showing the performance of the CNN (a) and the SVC (b) on their corresponding test datasets. The performance metrics are defined in Equation 1, the values between brackets for the SVC show the performance when the problem is considered as a binary problem; i.e. when 'Artefact' and 'Empty' are combined as 'No Plume'.

The trained CNN is applied to all 2020 data. Processing of the 5193 orbits of 2020 resulted in 752,890 scenes (only taking into account scenes with >20% valid pixels), of which 25,626 scenes (3.4%) are identified as containing plume-like morphological structures by the CNN. This number does include artefacts and duplicates due to the moving window algorithm, these are filtered later on. We use a subset of these scenes to train the second step of the machine learning pipeline (section 2.4).





## 2.3 Feature Engineering

Due to the difficult nature of methane retrievals, not every plume-like morphological structure in the $XCH_4$ field is an actual methane plume. Different types of surface variability and atmospheric or meteorological conditions are known to affect the retrieval (Lorente et al., 2021); if there is a strong correlation between the methane enhancement and retrieval parameters, e.g. the surface albedo or the aerosol scattering coefficient, the retrieved methane enhancement might be caused by the albedo or aerosol variation and could therefore be a retrieval artefact. Other common examples of artefacts are those on the borders of

clouds and coastlines or when the direction of the enhanced structure is not in agreement with the windfield but with surface structures instead.

  In order to automate the necessary further inspection we compute numerical values for several "features" of potential plumes through feature engineering. Feature engineering is a commonly applied approach in machine learning problems and is espe-

cially helpful when limited amounts of labeled training data are available. These features are a representation of information a human expert would use to inspect the potential plumes in order to determine whether a scene contains a genuine plume or an artefact. We construct feature vectors consisting of features based on the corresponding scene. These vectors are then used to train the second model of the machine learning pipeline, the SVC. An overview of all developed features is presented in Table C1.


  Fundamental to many of those features is "masking" the plume in order to isolate the plume pixels from the background. For this purpose we use which part of the scene that has triggered the CNN detection. The Class Activation Map (CAM) can be used to visualize the localized activations of a CNN corresponding to a certain class on which it was trained (Zhou et al., 2015). We apply Grad-CAM (Selvaraju et al., 2020), which allows the computation of the CAM for our CNN that includes

fully-connected layers. In our binary classification problem, the CAM visualizes which regions of the deepest feature maps (the 8x8, deepest max-pooling layer in Figure 2) lead to an activation of the plume class. This spatial activation is calculated using the gradients between all internal 64 feature maps and the fully-connected layer. In order to obtain a CAM of sufficient resolution we limit the depth of the CNN to two convolutional blocks. The CAM is upsampled to match the input resolution (Selvaraju et al., 2020). Figure 4b shows that the CAM correctly identified the plume-like structure in the XCH4 scene in

Figure 4a , disregarding noisy high-enhancement pixels elsewhere in the scene.





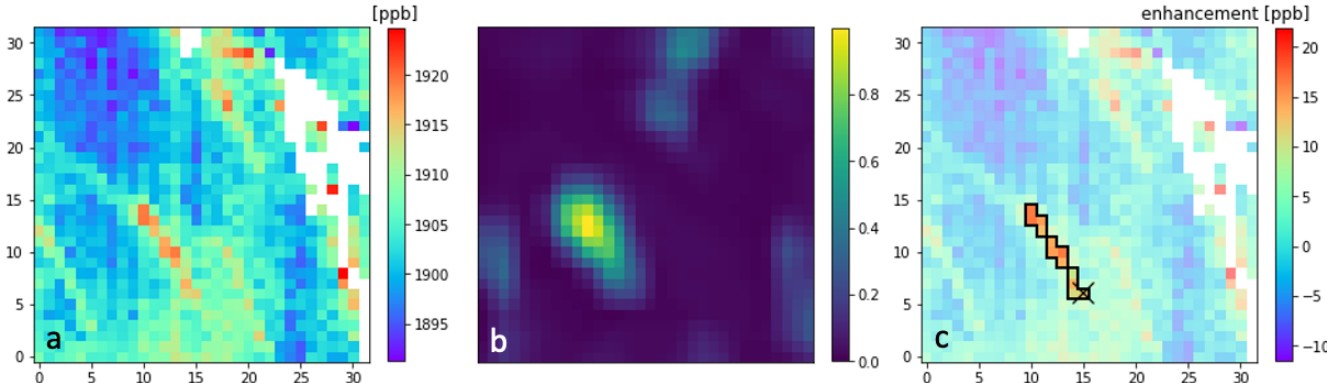

**Figure 4.** Several Feature Engineering results computed for the 32x32 scene from Figure 1. (a) The atmospheric methane mixing ratio XCH$_4$, (b) Class Activation Map which highlights the areas identified by the CNN as plume-like structures based on Figure 1c, the pre-processed scene, (c) methane enhancements relative to the local background with the black line indicating the high-confidence plume mask. The pixel with the 'x' is identified as the pixel most likely to contain the source location based on the plume mask and local windfield (shown in Figure 1a).

In addition to using the CAM to analyze CNN performance, we also use it to generate a binary plume mask. We multiply the CAM with the enhancement above the mean XCH$_4$ value minus the scene's standard deviation. The output is a map highlighting pixels with high methane enhancements that are identified as part of the plume by the CNN; we identify the pixel with the maximum value as starting point for the plume mask. To compute a "high confidence" plume mask, we start from the corre-

sponding pixel in the xch4 image and dilate outwards (including diagonally). We only add adjacent pixels with enhancements of 1.8 standard deviations above the mean (Figure 4c). We repeat this process with a lower threshold of 0.8 standard deviations to also obtain a "low confidence mask"; both thresholds were established empirically. This approach ensures that noise in other parts of the image is excluded from the plume mask. Several statistics of the (potential) plume can be computed using these masks with supporting data.


One of the major indicators of an artefact is a strong correlation with one or more retrieval parameters. If an enhancement in the XCH$_4$ field is caused by a surface (albedo) feature or by (enhanced) scattering in the atmosphere which is represented by the aerosol optical thickness, we expect their spatial patterns to be similar. Therefore, we calculate the correlation between XCH$_4$, and the surface albedo (SWIR), aerosol optical thickness, $\chi^2$ (an indicator for retrieval fit quality) and surface pressure

across the plume mask. We calculate these correlations for the high and low confidence plume masks as well as 1- and 2-times dilated versions of the low confidence mask, and the entire scene. We account for pixels outside the plume mask as we would expect a strong correlation around the enhancement if it is an artefact. The correlations over the entire scene reflect large scale patterns that do not necessarily imply artefacts.





Another major indicator for artefacts is a mismatch between the direction of the plume and the direction of the local 10-m
wind field from the ERA5 reanalysis (Hersbach et al., 2020) included in the TROPOMI Level 2 data product (Hasekamp et al.,
2022). By applying a Principle Component Analysis (PCA) we compute the two main axes of the pixels in the high confidence
plume mask, after re-projecting the pixel centers to meter space and weighting them by their enhancement relative to the
background. We use the ratio of the variances described by the axes as a measure of the plume's elongation. For elongated
plumes (e.g. Figure 4), the primary axis' variance is much larger than the variance projected to the secondary axis, while for
less elongated "blob"-like plumes this ratio is small. Furthermore, we compare the angle of the primary axis of the potential
plume to the angle of the wind direction (averaged across the plume mask), the smaller the difference, the more confidence we
have in the plume following the wind. We also use the wind field to identify the pixel that most likely contains the plume's
source by taking the 'most upwind' pixel within the high confidence plume mask (Figure 4c).

### 2.3.1   Source rate quantification

To estimate the source rates of the plumes observed by TROPOMI, we apply the Integrated Mass Enhancement (IME) method
(Frankenberg et al., 2016; Varon et al., 2018). This method relates the emission rate ($Q$) to the observed methane enhancement
in the plume (IME) and the local windfield (Varon et al., 2018):

$$Q = \frac{1}{\tau}\text{IME} = \frac{U_{\text{eff}}}{L}\text{IME} \quad ; \quad \text{IME} = \sum_{j=1}^{N} \Delta\Omega_j A_j \qquad (2)$$

where $\Delta\Omega_j$ denotes the methane column mass enhancement above the local background of pixel $j$ with footprint $A_j$. The
local background is calculated as the median value of the scene's pixels outside the high-confidence plume mask. The IME of
all $N$ pixels in the plume is related to the source rate using the average residence time $\tau$ of methane particles in the plume,
with $\tau$ given by the ratio between plume length $L$ and the effective wind speed $U_{\text{eff}}$. The plume length $L$ is approximated as
$L = \sqrt{A_M}$, where $A_M$ is the area of the plume mask (Varon et al., 2018). $U_{\text{eff}}$ can be expressed as a function of the local
(reanalysis) wind speed. Frankenberg et al. (2016) and Varon et al. (2018) developed the IME method for high resolution
instruments, for which the 10-meter winds are most representative for $U_{\text{eff}}$. For the larger scale of TROPOMI plumes, both
10-meter as well as boundary layer average (PBL) winds have been used (Varon et al., 2019; Schneising et al., 2020; Cusworth
et al., 2021; Tu et al., 2022a). As the most representative wind can vary from case to case, we use the mean of quantifications
using ERA5 10-meter winds (Hersbach et al., 2020), GEOS-FP 10-meter winds and GEOS-FP PBL winds (Molod et al., 2012).

We calibrate the relation between $U_{\text{eff}}$ and these local windspeeds by quantifying 15,336 plumes simulated with the Weather
Research and Forecasting model coupled with a Chemistry module, version 4.1.5 (Skamarock et al., 2019; Grell et al., 2005).
The model uses 38 vertical levels and three nested domains at a horizontal resolution of up to 4x4 km². Physical parameteri-
sations and meteorological initial and boundary conditions are as described in Dekker et al. (2017). We release passive tracers
with emission rates mostly between 10-100 t h⁻¹ at various locations in West Asia, Mexico, and Argentina for June-September





2019 and 2020. We sample the plume at the TROPOMI overpass time and quantify them as described above.

Using the model wind speeds and known emission rates, we find that $U_{\text{eff}}$'s dependence on the PBL wind is best described by a linear relationship: $U_{\text{eff}} = \alpha_1 \cdot U_{\text{PBL}} + \alpha_2$, with $\alpha_1 = 0.47$ and $\alpha_2 = 0.31$ ($r^2 = 0.78$). For the dependence on $U_{10}$, we also

find a linear relation as optimal and constrain $\alpha_2$ to be non-negative: $U_{\text{eff}} = \alpha_1 \cdot U_{10} + \alpha_2$, with $\alpha_1 = 0.59$ and $\alpha_2 = 0.00$ ($r^2 = 0.77$). We use the mean of the three resulting $U_{\text{eff}}$ values to quantify emissions.

To estimate uncertainty, we create an ensemble of estimates by varying the parameters influencing the quantification. For each of the different input wind speeds: we vary the threshold for masking the plume from 1.3 to 2.3 standard deviations (step

0.1); adjust the background concentration by $\pm 2$ times the mean XCH$_4$ uncertainty in the scene (step 0.4); vary the wind values from -50% to +50% (step 10%); and vary $\alpha_1$ and $\alpha_2$ for -5% to 5% (step 1%). We report the standard deviation of the resulting 43,923 member ensembles as uncertainty for each plume.





## 2.4 Support Vector Classifier (SVC)

A Support Vector Classifier (SVC) constructs hyperplanes as the optimal decision boundary to separate multiple classes in
high-dimensional feature space. SVCs in general perform better with datasets of limited size compared to deep learning al-
gorithms and are in general less prone to erroneous influence from outliers. We use 843 labeled scenes from 2020 classified
by the CNN to contain plume-like structures as a training dataset for the Support Vector Classifier (SVC). About half of the
scenes are randomly selected from within 7 geographical zones with specific types of predominant artefacts and the other half
is selected randomly. Scenes are labeled as 'plume' (444 scenes), 'artefact' (341 scenes) or 'empty' (58 scenes, indicating there
is not a clear plume or artefact). We have only included scenes with unambiguous labels. The fact that there are relatively few
'empty' scenes in this subset indicates the CNN performs well. We use balanced class weights to correct the imbalance in the
number of training samples per class, the weights are inversely proportional to the number of scenes in each class.

We train the SVC on a vector of 41 features per scene including: correlations with retrieval parameters for different plume
masks, the angle between the wind and elongated direction of the plume, the elongation ratio of the plume, the source rate, and
several statistical properties (all engineered features are listed in Table C1). We do not include features such as latitude and
longitude or distance to a known source or known infrastructure in order to be unbiased to where a scene is located. We train
the SVC to find the optimal classification boundary within this 41 dimensional space based on the 843 labeled feature vectors.
Each feature is standardized by subtracting the mean, and scaling the value to the unit variance of that feature in the entire
trainingset. We use a Radial Basis Function (RBF) kernel and set the regularization parameter to 1.2, this hyperparameter was
optimized using a simple grid search optimization. The gamma value is scaled inversely to the number of features (41 in this
case) multiplied with the variance of the training dataset, as is common practice and helps to homogenize the features which
have different units and ranges of values (Table C1).

We randomly split the labeled dataset into a training set (80%) and a test set (20%). Contrary to the CNN, when training an
SVC no validation set is used. Correctly detecting plumes is of predominant interest, therefore we combine non-plume scenes
(artefact & empty classes) when evaluating the performance. We train the model 2000 times for different splits of the dataset.
The distribution of these different realizations shows convergence with binary kappa = 0.78 ± 0.04 and recall = 0.88 ± 0.03
(standard deviations), indicating the model setup is not too dependent on the data split. We select a model with relatively high
kappa and recall, where performance is similar between the training and test sets. Figure 3b shows the 3-class performance
on the test set, indicating distinguishing between plumes and artefacts is the most challenging distinction for the SVC. The
binary Cohen's Kappa score of the selected model is 0.83. A Kappa score of above 0.8 is generally seen as a good classification
performance. The recall is 0.93, meaning that 93% of the scenes with plumes which were present in the test set are successfully
identified, and only 7% of the plumes are missed.



To evaluate which features are important for the SVC to classify a scene, we performed a permutation importance analysis perturbing each feature 40 times (Breiman, 2001). Based on the resulting mean feature importance, the most important features are: correlation of $XCH_4$ with $\chi^2$, the CNN score, the albedo correlation, the enhancement of the plume, the fraction of valid pixels, the angle with the local wind, and the average quality flag of plume pixels (the top 10 ranking feature importance metrics are presented in Table C1). These correspond to what is important to a human expert labeler.

## 2.5 Plume characterization

Due to the moving-window algorithm, each group of 16x16 pixels is seen by the ML pipeline up to four times. This ensures plumes do not go undetected because they are cut in half and allows multiple close by plumes to be detected in adjacent scenes, but also leads to duplicate detections. Therefore, plumes for which the generated plume mask overlaps with a plume mask from another scene are grouped into a group. For each group, the scene with the highest IME value is selected.

To assess which anthropogenic activity might underlie a detected plume, we use the estimated source location to find the local dominant source type in gridded bottom-up inventories. We exclude sectors that are unlikely to produce point-source emission signals in the TROPOMI, such as rice cultivation and livestock. We include 2019 oil, gas and coal emissions from the Updated Global Fuel Exploitation Inventory (GFEI v2) (Scarpelli et al., 2022) and 2018 landfill emissions from EDGAR V6.0 (Crippa et al., 2021). We identify the dominant source type in a $0.7°\text{x}0.7°$ square centered around the estimated source location. Based on known emitters, we found that using a window of this size mitigates errors in the estimated source location as well as spatial errors in the emission inventories. We do not use this approach to attribute detections to wetlands. However, we do inspect 2019 fluxes from WetCHARTs v1.3.1 ensemble (Bloom et al., 2021) to identify regions where detections might be influenced by strong wetlands fluxes, such as central Africa (Pandey et al., 2021).

In order to test the automated pipeline and feature engineering algorithms, we apply it to data over a September 2019 super-emitter event in Louisiana, US (Maasakkers et al., 2022a). The model detects the emission event on multiple days, including on the first day with large emissions and significant TROPOMI coverage (September 25). The CNN score is >0.999 and the SVC classified the scene as a plume. The estimated source location of the plume is 2.2 km away from the source and our automated quantification estimate is $121 \pm 46$ t h$^{-1}$, in good agreement with the inversion-based quantification by Maasakkers et al. (2022a) of 101 (49-127) t h$^{-1}$.





### 2.6 High spatial resolution methane satellite instruments, retrievals and source rate quantification


We use observations from three high spatial resolution instruments, GHGSat, PRISMA and Sentinel-2 to inspect the sources of the detected methane plumes. This section describes the main characteristics of these instruments.

#### 2.6.1 GHGSat

GHGSat-Cx instruments are Fabry-Perot imaging spectrometers launched in 2020-2022 (C1-C5) building on the GHGSat-D

instrument (Jervis et al., 2021). The instruments have a spatial resolution of 25x25 $m^2$ over targeted scenes of ∼10x15 $km^2$ (Ramier et al., 2020; MacLean et al., 2021). They sample the SWIR part of the spectrum between 1630 and 1675 nm at ∼0.3 nm spectral resolution, retrieving methane column density with a precision of 1% of the background concentration and a theoretical detection threshold of down to ∼100 kg/h at a windspeed of 3 m s$^{-1}$ (ESA, GHGSat, 2022). During a controlled release experiment (Sherwin et al., 2022) a plume of ∼200 kg/h was successfully detected. We use data from GHGSat-C1 and

C2 and estimate source rates using the IME method as described in Maasakkers et al. (2022b) for point sources with $U_{10}$ winds from GEOS-FP (Molod et al., 2012). The uncertainty in the quantification is estimated as described by Varon et al. (2019), taking into account error contributions from measurement noise, the wind speed and the IME method similar to Maasakkers et al. (2022b).

#### 2.6.2 PRISMA

The Italian Space Agency's hyperspectral instrument PRISMA was launched in March 2019 and generates publicly available targeted hyperspectral 30x30 $km^2$ images at a spatial resolution of 30x30 m and ∼10 nm spectral resolution (Cogliati et al., 2021; Guanter et al., 2021; Cusworth et al., 2021). The minimum revisit time can be up to 7 days with ±20% across-track pointing. The smallest source rate tested during a controlled release experiment (Sherwin et al., 2022) is ∼2500 kg/h. The theoretical detection threshold is lower (∼300-900 kg/hr for homogeneous scenes) and strongly depends on the surface type/homogeneity

(Guanter et al., 2021). The PRISMA instrument is not a continuous mapper, but a data archive of past (targeted) observations is publicly available. PRISMA can also be used to target a location of interest on a future moment in time. We perform methane retrievals and IME quantifications as described in Guanter et al. (2021) using plume masking following Varon et al. (2018) and GEOS-FP 10m wind data (Molod et al., 2012).

#### 2.6.3 Sentinel-2

The Sentinel-2 surface-imaging mission (consisting of Sentinel-2A launched in 2015 and Sentinel-2B launched in 2017) was demonstrated by Varon et al. (2021) to be capable of the detection of methane super-emitter plumes under favorable conditions. Both satellites carry a MultiSpectral Instrument, with a pixel resolution of 20 m for the B11 (∼100 nm) and B12 (∼200 nm) SWIR bands with sensitivity to methane. The instruments have a 290 km wide swath resulting in a global 2-5 day revisit time (Drusch et al., 2012), Sentinel-2 observes continuously (as opposed to GHGSat and PRISMA) and provides an extensive

publicly available archive going back years. A methane absorption signal has to be strong in order to stand out within the





aggregated signal of the entire band, therefore only relatively large quantities of methane can be retrieved and the detection limit worsens considerably over non-homogeneous terrain. The detection limit is estimated at $\sim$1-2 t h$^{-1}$ for homogeneous scenes (Gorroño et al., 2022), in agreement with Sherwin et al. (2022) where a $\sim$1800 kg/h emission was detected and quantified. We apply the methane retrieval and IME quantification approach from Gorroño et al. (2022) that similarly to Varon et al. (2021)

uses a "reference day" without a plume to isolate the difference caused by methane concentration enhancement. We again use GEOS-FP 10m wind data (Molod et al., 2012).





# 3 Results

We apply the trained and optimized CNN and SVC models in sequence on all 2021 TROPOMI XCH$_4$ data. Analysing the full year with the ML pipeline takes approximately three hours on a single core. From the 794,395 (32x32 pixel) scenes, the CNN identifies 26,444 scenes (3.3 %) as containing plume-like XCH$_4$ morphological structures. The SVC classifies 10,430 of these scenes as plumes. After duplicate removal, 4869 scenes are identified as unique. These 4869 scenes are manually inspected to assess the performance of the pipeline. We confirm 2974 scenes as confident plumes. Another 745 scenes are labeled as 'potential plumes', accepting these scenes as plumes results in a precision of 76% for the full pipeline. These potential plumes could not readily be verified as real methane plumes, but are valuable for further inspection. The remaining scenes are either labeled as artefacts or not containing a (concentrated) plume. These misclassifications can be used to further optimize the machine learning pipeline. Here, we will focus on the 2974 'confident plumes', and present the result of our high-resolution satellite instrument analysis to pinpoint the exact sources of twelve (clusters of) detections.

## 3.1 Overview of the confirmed detections in 2021

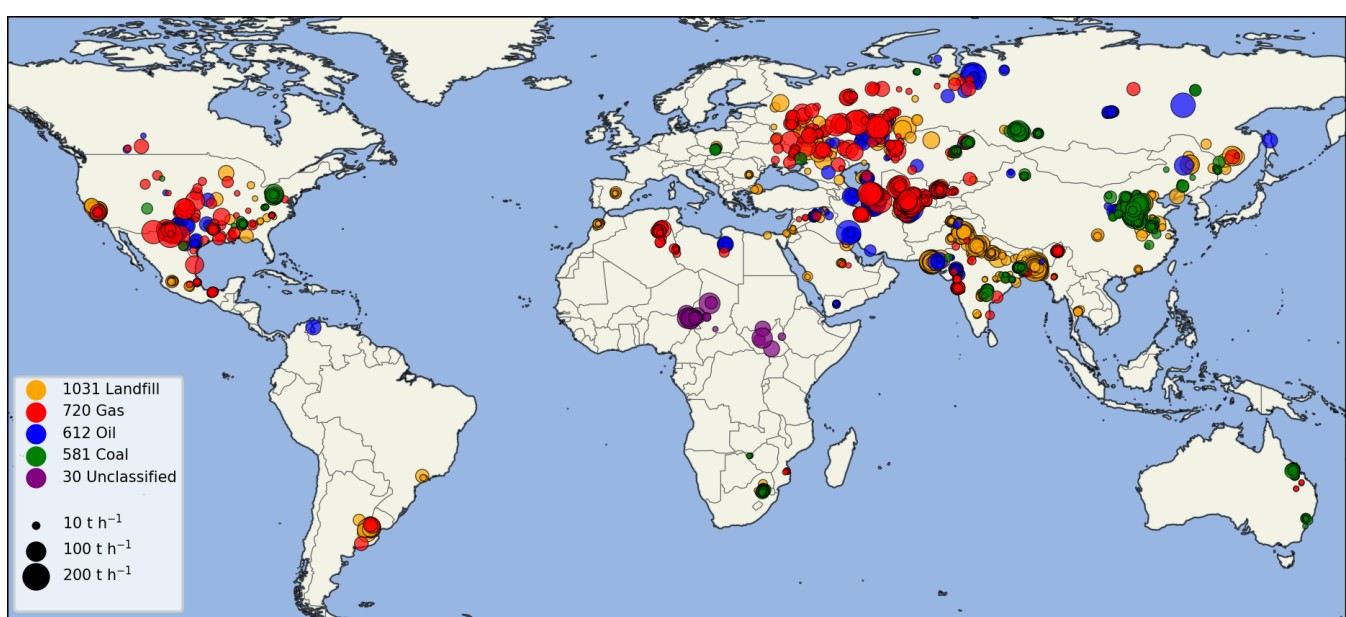

**Figure 5.** All 2974 confident plume detections for 2021, grouped into one of the four dominant anthropogenic source types and sized by source rate. 30 detections in central Africa are labeled as 'Unclassified'.



Figure 5 shows the spatial distribution of all 2974 detected and confirmed 2021 plumes, classified by source sector based on the three bottom-up inventories (Section 2.5). We find that 1031 plumes predominantly relate to urban areas/landfills, 720 to gas infrastructure, 612 to oil infrastructure and 581 to coal mines. As super-emitters are usually not the result of regular operations, and are therefore not well represented in bottom-up inventories, especially large transient emissions may be misattributed by this approach. Wetlands are not expected to result in point source emissions, but strongly emitting wetlands in central Africa,

such as South-Sudan and the Niger delta, can produce large enhancements in TROPOMI data (Pandey et al., 2021; Shaw et al., 2022). In the absence of large anthropogenic emissions, we label plumes from these two regions as 'unclassified'. Wetlands might also contribute to detected signals in areas with large anthropogenic emissions like the Dhaka delta (Bangladesh) and the Mississippi delta (US).

There are many clear hot spot locations with frequent detections. To group the detections into clusters with a common source, we apply the DBSCAN clustering algorithm (Ester et al., 1996; Schubert et al., 2017). We cluster based on the distance between detections in meters and set a threshold of 5 detections within 30 km as the minimum to identify a cluster. We identify 94 clusters, this is a conservative estimate for the number of persistent locations as some known persistent emitters have fewer than 5 detections in 2021. We also observe several areas with extensive plumes from multiple emitters such as the west-coast

of Turkmenistan which are grouped into one big cluster. We find the majority of detected plumes (74.8%) to be clustered at a persistent urban or fossil fuel exploitation source and classify the remaining plumes as transients. Zoom-ins of the clusters in several distinct regions as well as source rates for all detections are shown in Figure 6.

**Figure 6.** Regional plume detections showing color-coded persistent emission clusters with transient emissions shown in black. (a) Large clusters of detections related to oil/gas exploitation in Turkmenistan. (b) The clearly distinguishable outlines of the Delaware Basin and Midland Sub-Basins within the Permian Basin, US. (c) Detections show the same spatial structure along compressor stations and pipelines in western Russia. (d) Clusters of hotspots in eastern China with the extensive Shanxi coal-mining region in the center. (e) Clusters of coal mining detections in north-east Australia. (f) A clear cluster of detections around the persistent source in Casablanca, Morocco. (g) The distribution of estimated source rates for all 2974 detected plumes in the year 2021, capped at 200 t h$^{-1}$. The 5$^{th}$ and 95$^{th}$ percentile as well as the mean values of the distribution are shown as vertical lines.





Several of the identified clusters are located over well-known oil/gas production regions such as the west coast of Turk-
menistan (Figure 6a) previously studied by Varon et al. (2019) and Irakulis-Loitxate et al. (2022b), Algeria (Varon et al.,
2021), Libya, and multiple basins in the US (Shen et al., 2022) including the Eagle Ford, Haynesville, and most prominently
the Permian (Figure 6b) where Zhang et al. (2020) quantified emissions based on TROPOMI and we find individual clusters of
detections over the Midland and Delaware sub-basins. The fact that many of the detections are clustered around known large
sources gives confidence in the performance of the models that did not use prior location information. We also identify oil/gas
production clusters which have not been studied in detail, such as in northern Libya, Yemen, and northeastern India.

We also find large transient plumes along the major gas transmission pipelines in western Russia (Figure 6c), similar to what
Lauvaux et al. (2022) found for 2019-2020. Clusters of detections are seen over coal mining areas in China (Chen et al., 2022),
southern Poland (Tu et al., 2022b), South Africa, Russia, and northeastern Australia (Figure 6e) where Sadavarte et al. (2021)
quantified large emissions from these clusters of coal mines. Our approach allows us to detect which specific locations within
a larger area of fossil fuel exploitation cause large methane plumes, examples are the super-emitter clusters within the large,
spread-out Shanxi coal-mining region in China (Figure 6d).

The majority of our detections are related to urban areas around the world, including four cities with large fluxes (Buenos
Aires, Mumbai, Delhi and Lahore), which were also identified by Maasakkers et al. (2022b) based on long-term wind-rotated
TROPOMI averages. Urban areas comprise a range of source types but individual landfills can make up a large fraction of
total urban emissions (Maasakkers et al., 2022b). When we zoom into the area around Casablanca, Morocco (Figure 6f), we
see strong convergence in a cluster. Most plumes within the cluster (19 out of 23) are quantified below 25 t h$^{-1}$, of which 8
are quantified below 15 t h$^{-1}$. The estimated source locations of the plumes are on average 12 km away from a landfill later
detected and quantified using GHGSat observations (Figure 7, Section 3.2). Other urban clusters include Madrid in Spain; 7
cities in Pakistan; Riyadh in Saudi Arabia; Bucharest in Romania; and Mexico City and Guadalajara in Mexico. The most
frequently detected (104 detections) urban cluster is centered around Dhaka, Bangladesh. In India, we see 8 urban clusters and
several cities with at least two detections. Detections over India are seasonally limited by meteorology as there are hardly any
TROPOMI data during the May-September monsoon season because of persistent cloud cover.


The distribution of the estimated source rates of all 2974 verified plumes is shown in Figure 6g. Our IME-based quantifi-
cations show mean emissions of 44 t h$^{-1}$, with a large 5-95$^{th}$ percentile range of 8-122 t h$^{-1}$. Many detections are quantified
below the detection threshold of previous TROPOMI plume identification & quantification methods of 25 t h$^{-1}$ (Lauvaux et al.,
2022; Jacob et al., 2022). We find 1143 plumes quantified under 25 t h$^{-1}$ including 241 plumes under 10 th$^{-1}$. Many of these
originate from persistent emission clusters where emissions have been confirmed using high-resolution instruments. Although
the applied mass balance quantification method has significant uncertainty, this shows the plume detection limit of TROPOMI
is better than previously reported in literature.





In order to present a rough estimate for the total emissions represented by the detected plumes, we assume each emission
event is active for 24 hours (the minimum sampling frequency of TROPOMI). For some transient plumes, such as pulse emissions at compressor stations, the 24 hours estimate can be an overestimate, but we take these as representative for similar
transient events occurring outside of the TROPOMI observation window. Using these assumptions, we find detected emissions
of 3.1 ±1.3 Tg for 2021. As a conservative uncertainty estimate we use the sum of the standard deviations of the individual
ensembles. The number of detected plumes is an underestimate of the true number of plumes as observations are limited by
clouds and illumination.

To account for the limited TROPOMI coverage and get an indication of the annual emissions our detections are representative
of, we scale our detected emissions by the fraction of days with coverage. We estimate the local number of days with coverage
from our 794,395 valid scenes by first mapping their spatial footprints to a 0.1x0.1 degree grid, removing duplicate coverage
from overlapping scenes in the same orbit. We then correct for local variations in coverage (such as persistent areas without
data) by convoluting this field with the summed footprints of all 2021 TROPOMI data at 0.1x0.1 degrees. We finally aggregate
our detected emissions to a 1x1 degree grid and divide those by the fraction of days in 2021 with coverage resulting from the
coverage map average to 1x1 degrees. We find a scaled-up annual emission flux of 10.3 Tg, which is approximately 2.7% of
total bottom-up 2017 anthropogenic emissions (380 Tg yr$^{-1}$) (Saunois et al., 2020). Super-emitter plumes from landfills account
for 4.1 Tg yr$^{-1}$, those from coal 2.1 Tg yr$^{-1}$, those from oil 2.2 Tg yr$^{-1}$ and those from gas 1.9 Tg yr$^{-1}$ each. These estimates
are only small fractions of total anthropogenic emissions as our conservative upsaling approach only takes large TROPOMI-
detected super-emitter plumes into account. Emissions from smaller point sources and area sources make a large contribution
to the total and are better captured by an atmospheric inversion.





### 3.2 Synergy of automated TROPOMI detections with high-resolution instruments

We use the detection of persistent methane plumes in TROPOMI data to target high-resolution observations (GHGSat-C1/C2) and data analysis (PRISMA archive, Sentinel-2) following (Maasakkers et al., 2022b). Furthermore, we investigate large transient emissions with data from non-targeted instruments such as Sentinel-2.

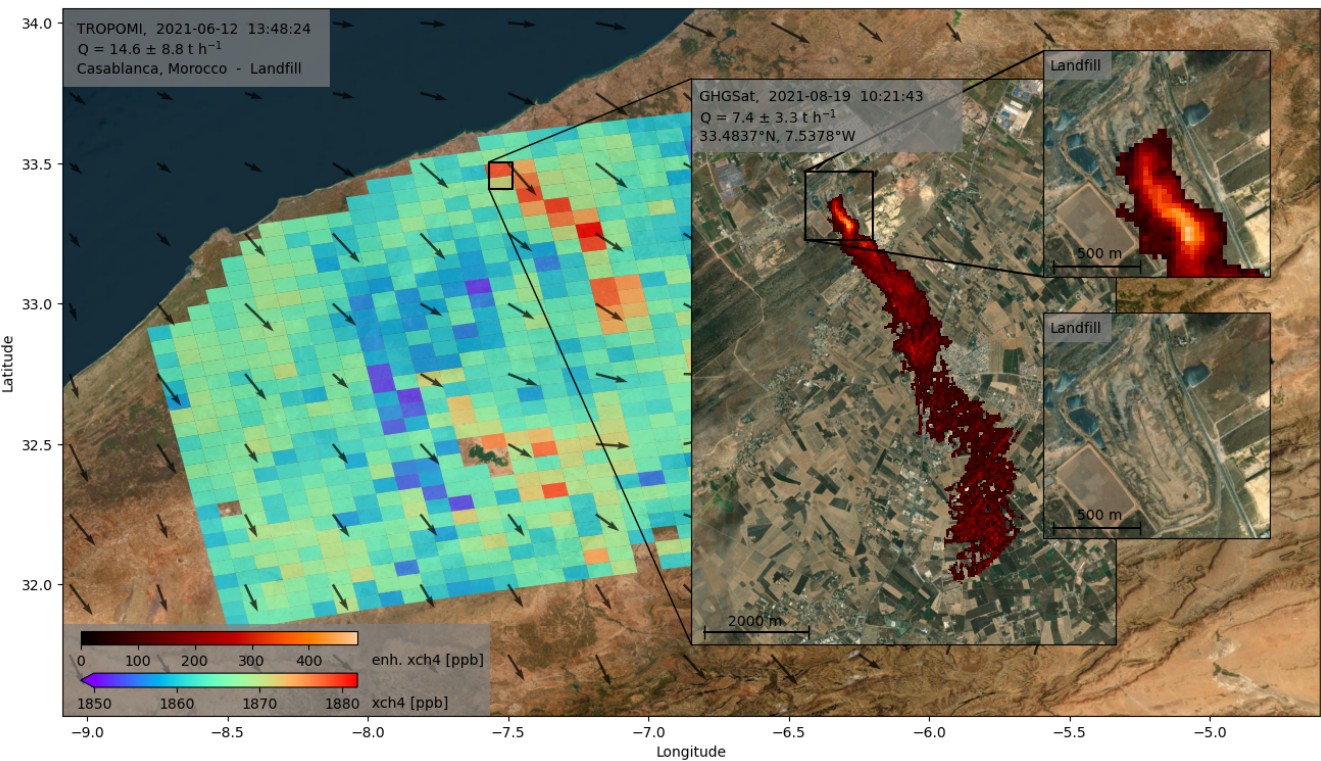

**Figure 7.** Methane plumes detected from Casablanca, Morocco on two different days, with TROPOMI and GHGSat data overlaid over visual ESRI World Imagery (Esri, Maxar, Earthstar Geographics, and the GIS User Community, 2022). Timestamps are UTC. The plume observed by TROPOMI on 2021-06-12 is quantified at $14.6 \pm 8.8$ t h$^{-1}$. The plume observed by GHGSat on 2021-08-19 originates from the landfill between Casablanca and Médiouna (33.483, -7.538) and is quantified at $7.4 \pm 3.3$ t h$^{-1}$. The winds are the GEOS-FP 10 meter windfield (Molod et al., 2012).

Figure 7 shows a TROPOMI plume ($14.6 \pm 8.8$ th$^{-1}$) detected near Casablanca in Morocco on June 12th, 2021. We detected 23 plumes in the area in 2021 with source rates ranging from $8.9 \pm 5.1$ t h$^{-1}$ to $40.5 \pm 18.0$ t h$^{-1}$ with a mean of 18.8 t h$^{-1}$,
indicating a persistent source (Figure 6f). Based on wind-rotation analysis (Maasakkers et al., 2022b), we find the landfill located in between Casablanca and Médiouna to be the optimal target for high-resolution observations. Based on this TROPOMI analysis, we "tip-and-cue" GHGSat to observe this location. The inset image shows a targeted GHGSat-C2 observation on August 19th, 2021, indeed showing a methane plume (quantified at $7.4 \pm 3.3$ t h$^{-1}$) originating from the landfill and extending





downwind.


Figure 8 shows two methane plumes detected with TROPOMI in northern Kazakhstan on May 14th, 2021 at 07:53 UTC, quantified at $35.2 \pm 13.2$ t h$^{-1}$ for the northern and $28.1 \pm 11.2$ t h$^{-1}$ for the southern plume. The same locations were also detected in an adjacent orbit at 09:33 UTC but the closest days with coverage before and after May 14 do not show emissions, which indicates the plumes are transient. The bottom-up inventories give natural gas systems as the locally dominant

anthropogenic source sector because of the presence of a gas transmission pipeline. In Sentinel-2 observations taken 38 minutes before the first detection, we find two emitting locations close to the pipeline in the upwind part of the TROPOMI plume masks. The source rates of the Sentinel-2 plumes are $180 \pm 59$ t h$^{-1}$ and $75 \pm 23$ t h$^{-1}$ for the northern and southern plume respectively. The rather large discrepancy between the TROPOMI and Sentinel-2 quantifications can be explained by the uncertain low wind speeds, the not well-developed plume in TROPOMI increasing the uncertainty in the IME, and possibly the

partial pixel enhancement effect described by Pandey et al. (2019).

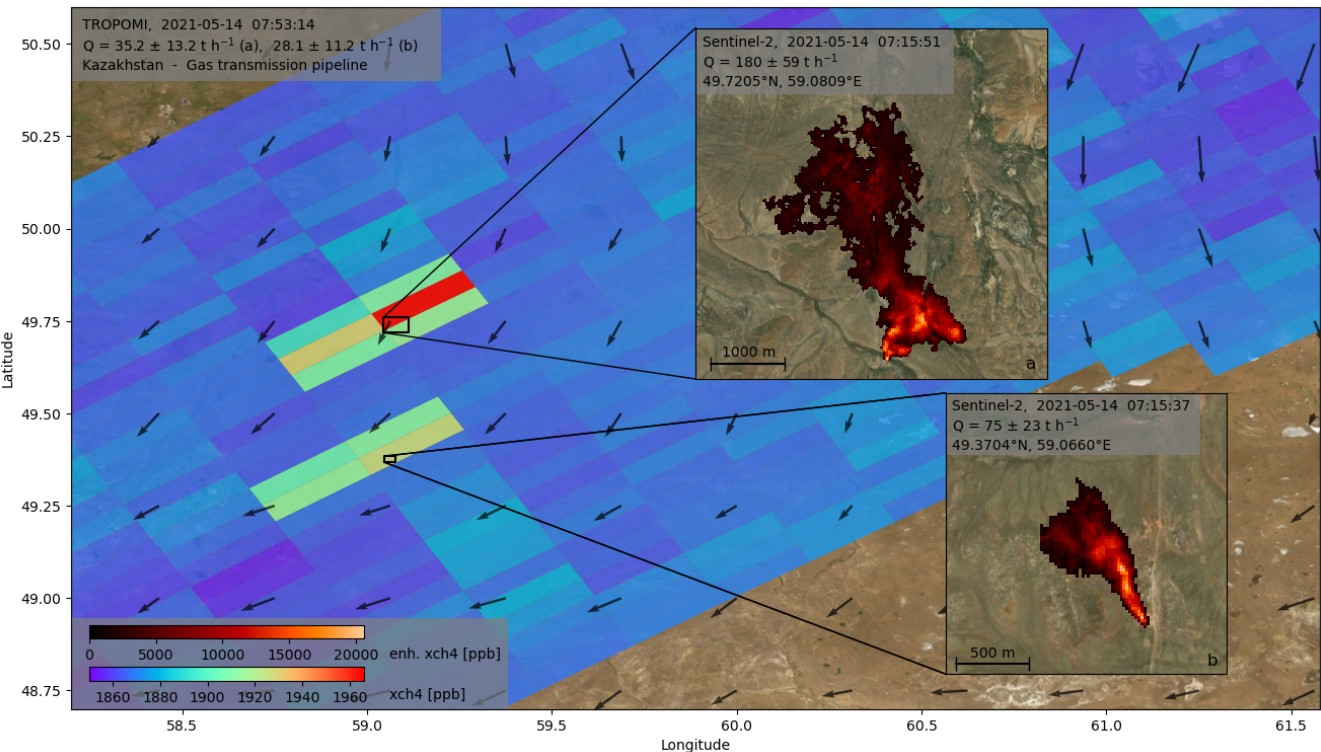

**Figure 8.** Transient methane plumes detected at two different locations in northern Kazakhstan with TROPOMI (quantified at $35.2 \pm 13.2$ t h$^{-1}$ for the northern and $28.1 \pm 11.2$ t h$^{-1}$ for the southern plume) and Sentinel-2 overlaid over visual ESRI World Imagery (Esri, Maxar, Earthstar Geographics, and the GIS User Community, 2022). Timestamps are UTC. The plumes originate from natural gas pipeline infrastructure. The winds are the GEOS-FP 10 meter windfield (Molod et al., 2012).





**Figure 9.** (Caption next page.)



**Figure 9.** (Previous page.) Plumes detected over 10 locations which were inspected with high-resolution instruments. Observations at the same location with different instruments are most often not on the same day. Details are provided in Tables B1-B3 for the high-resolution instruments and in Table B4 for TROPOMI. TROPOMI data is shown in Mercator projection (EPSG:4326), high-resolution data is shown in the local Universal Transverse Mercator (UTM) projection. The data is overlain over visual ESRI World Imagery (Esri, Maxar, Earthstar Geographics, and the GIS User Community, 2022). The worldmap in the center of the image corresponds to Figure 5, showing all 2974 detected plumes in 2021. TROPOMI data is here displayed as enhancement relative to the median $XCH_4$ of the 32x32 pixel scene. Several of the zoomed-in views with high-resolution data were set to an opacity of 0.5 in order to reveal the infrastructure at the source of the plume.

Figures 7 and 8 show how TROPOMI detections can be combined with high-resolution observations for both persistent and transient emitters. Figure 9 shows ten additional locations analyzed with GHGSat (7 scenes), PRISMA (2), and Sentinel-2 (1) based on TROPOMI detections. Supplemental Tables B1-B3 provide details on these high resolution observations and associated (not necessarily on the same day) TROPOMI scenes (Table B4). We find facility-level source rates from $0.3 \pm 0.1$ t h$^{-1}$ up to $16 \pm 5$ t h$^{-1}$. Most scenes contain multiple emitters that are so close together that they would be impossible to separate using only TROPOMI observations.

The different specifications of the high-resolution instruments make them suitable for different purposes. The methane designated GHGSat-Cx instruments have the lowest detection limit and are capable of retrieving methane over areas with challenging surface structures such as urban areas. Southeast of Madrid we observe plumes from two separate landfills located 7 km apart. The methane plumes originate from the active areas of the landfills where waste is added. In TROPOMI, the signals from these two landfills appear as a single point source. In the gas production region around Shreveport, Louisiana, US, we observe three emission plumes originating from distinct infrastructure including from two facilities that are only ∼500 m apart. Over a coal mining area in Russia we see in a GHGSat observation that a single TROPOMI-based target has contributions from 10 different point sources with source rates from $0.2 \pm 0.1$ to $2.4 \pm 1.1$ t h$^{-1}$. The emissions originate from coal mining facilities, such as underground mine vents, add up to 8.8 t h$^{-1}$. In the Assam oil and gas production region in India, we find five plumes adding up to 12.9 t h$^{-1}$, showing the GHGSat-Cx instruments are also capable of retrieving methane plumes over non-homogeneous areas including forest and agricultural lands. We also use GHGSat to target two less challenging desert scenes with oil and gas production. A GHGSat observation over Libya shows two sources downwind of each other, the most upwind source is an unlit flare stack. At the border of Uzbekistan and Turkmenistan (listed as Uzbekistan in Figure 9), we find emissions at three distinct locations within a single natural gas facility, with source rates ranging from $3.6 \pm 1.0$ to $6.4 \pm 1.8$ t h$^{-1}$. These emissions appear to originate from unlit flaring stacks, similar to what Irakulis-Loitxate et al. (2022b) and Varon et al. (2019) found for other natural gas facilities in the region.

For scenes with more homogeneous surfaces or extremely large emissions, PRISMA and Sentinel-2 can also be used. The PRISMA observation in Turkmenistan shows three plumes with an aggregated emission rate of $9.0 \pm 2.9$ t h$^{-1}$. The emissions originate from distinct pieces of gas infrastructure (quantified at $3.5 \pm 1.1$, $3.0 \pm 1.0$ and $2.5 \pm 0.8$ th$^{-1}$), the sources are





located within the footprint of a single TROPOMI pixel (the same source as shown in Figures 1 and 4). We also use the PRISMA archive to detect a plume originating from a coal mine ventilation shaft in Liuzhuang Village in Shanxi, China. Finally, we use Sentinel-2 to investigate a single location in Iran with complex observation conditions (elevation) where we only had a single TROPOMI detection in 2021. The emission therefore appeared to be transient at first, but with Sentinel-2 we find three emission plumes ranging from $1.3 \pm 0.4$ to $10.0 \pm 3.0$ t h$^{-1}$ originating from the same oil facility in a timespan of two months. Extensive monitoring of a location of interest over a longer timespan is feasible using Sentinel-2 (Varon et al., 2021; Irakulis-Loitxate et al., 2022b).





## 4 Conclusions

We detected methane emission plumes in 2021 TROPOMI data using an automated, machine-learning-based pipeline. We have trained a Convolutional Neural Network with a relatively small set of known plumes in pre-2021 TROPOMI methane data to detect plume-like morphological structures (kappa score of 0.97 and recall of 0.98 on the test set). We then used a Support Vector Classifier to distinguish real plumes from retrieval artefacts using additional information from the scene and supporting data (kappa score of 0.81 and recall of 0.93 on the test set). This two-step approach can also be applied to other instruments in the future. We tested our detection, source localization, and emission quantification estimate for a specific well-characterized natural gas well blowout and found it was accurately captured by our monitoring system. After application of our pipeline to 2021 data, we targeted high-resolution observations and analyses to find the facilities responsible for twelve (clusters of) plumes seen in TROPOMI.

Using our automated machine learning pipeline we scan all 794.395 scenes of 2021 in three hours on a single core. 4869 scenes are automatically classified as plume, of which 2974 are manually verified as confident plumes and 745 as potential plumes; giving the automated pipeline a precision of 61-76%. The most challenging distinction for the SVC is between plume and artefact, a distinction that even for a human expert can be inconclusive for difficult cases. We focus on the manually verified plumes, the remaining 39% of the scenes are mostly difficult to classify and can still be followed up with manual inspection or be used to further train the models. We find that most plumes (74.8%) originate from 94 clusters of detections around both known and new persistent source locations. The other plumes are mainly caused by transient emission events, such as along natural gas transmission pipelines in Russia. We most often detect plumes (based on bottom-up emission inventories) from urban areas / landfills (1031 plumes), followed by 720 plumes from gas infrastructure, 612 from oil infrastructure and 581 from coal mining. Many of the identified clusters are located at well-known fossil fuel exploitation regions, or urban areas known to emit methane. We also identify several previously unstudied sources such as in Libya and Assam (India) and identify specific super-emitting locations within spread-out fossil fuel production regions like the Shanxi coal mining area in China. Based on IME quantifications of all plumes, we found mean emissions of 44 t h$^{-1}$ with a 5-95$^{th}$ percentile range of 8-122 t h$^{-1}$, which is an indication of the TROPOMI detection limit. With 1143 detections under 25 t h$^{-1}$ including 241 plumes under 10 t h$^{-1}$, our automated approach has a better detection limit than previously published methods based on TROPOMI data. When we assume all 2944 detected anthropogenic emissions to be active for 24 hours, we find detected 2021 emissions of 3.1 ±1.3 Tg. Accounting for the limited coverage of TROPOMI, these detected emissions are representative of 10.3 Tg yr$^{-1}$, which is approximately 2.7% of global annual anthropogenic emissions.

For twelve locations, we used high-resolution satellite observations (GHGSat-C1/C2, PRISMA, and Sentinel-2) to identify the exact sources responsible for the detected plumes in TROPOMI. We utilized the different strengths of the high-resolution instruments; we made targeted observations with GHGSat over scenes with complex surface reflectance, whereas the archive of Sentinel-2 is used to analyze large transient emission events and track intermittent emissions. We found point sources from





landfills and fossil fuel exploitation with emission rates from $0.3 \pm 0.14$ t h$^{-1}$ to $180 \pm 59$ t h$^{-1}$. Most fossil-fuel-related

TROPOMI plumes had contributions from multiple point sources, with one GHGSat observation over Russia revealing emissions from ten different sources.

    Over the next few years, the number of global, regional, and point-source mapping instruments capable of retrieving methane plumes will vastly increase; with i.a. Sentinel-5, CO2M, MethaneSAT and Carbon Mapper (Jacob et al., 2022). Our monitor-

ing system can incorporate these fast-growing data volumes and can already be used to automatically detect plumes in the operational TROPOMI data and "tip-and-cue" high-resolution satellite instruments to find the associated super-emitting facilities. This identification and monitoring of super-emitters with large mitigation potential is paramount to reach the goals of the Global Methane Pledge.





*Code and data availability.* TROPOMI data are publicly available at https://ftp.sron.nl/open-access-data-2/TROPOMI/tropomi/ch4/18_17/.

GHGSat-C1/C2 methane plume data used in this study are available at https://doi.org/10.7910/DVN/QQQ9IU. Sentinel-2 data are publicly available at the Copernicus Open Access Hub https://scihub.copernicus.eu/. PRISMA data are publicly available at https://prisma.asi.it/, after registering at https://prismauserregistration.asi.it. GEOS-FP wind data can be downloaded at https://gmao.gsfc.nasa.gov/GMAO_products/. ERA5 wind data are available at https://cds.climate.copernicus.eu. WRF-CHEM code is available at https://github.com/wrf-model/WRF/releases, in this work version 4.1.5 was used. The GFEI (v2) emission inventory

is available at https://doi.org/10.7910/DVN/HH4EUM. The WetCHARTs emission inventory is available at https://daac.ornl.gov/cgi-bin/dsviewer.pl?ds_id=1502. EDGAR v6 data is available at http://data.europa.eu/89h/97a67d67-c62e-4826-b873-9d972c4f670b. The dataset of detected plumes in 2021 TROPOMI data will be made available upon publication.





## Appendix A: CNN trainingdata table

|  | split | subsplit | Nr of scenes | Nr of scenes augmented |
|---|---|---|---|---|
| **All labeled scenes** | 1.0 |  | 3070 |  |
| training set | 0.8 |  | 2456 | 19648 |
| - training subset |  | 0.8 | - 1965 | - 15718 |
| - validation subset |  | 0.2 | - 491 | - 3930 |
| test set | 0.2 |  | 614 | 4912 |

**Table A1.** An overview of the split in training and test data for the CNN.





**Appendix B:** **Details on the plumes observed with high resolution instruments**

| **GHGSat** | | Observation time [UTC] | Latitude [deg] | Longitude [deg] | Source rate [t h$^{-1}$] | Wind speed $U_{10}$ [m s$^{-1}$] | Sector |
|---|---|---|---|---|---|---|---|
| Casablanca, Morocco | | 2021-08-19 10:21:43 | 33.4837 | -7.5378 | 7.4 ±3.3 | 2.6 | Landfill |
| Madrid, Spain | a | 2021-12-13 10:04:26 | 40.3222 | -3.5913 | 4.3 ±2.0 | 2.7 | Landfill |
| | b | 2021-12-13 10:04:26 | 40.2611 | -3.6364 | 5.6 ±2.5 | 2.7 | Landfill |
| Libya | a | 2021-04-17 08:30:12 | 28.9089 | 20.9807 | 8.8 ±2.0 | 8.6 | Oil exploitation |
| | b | 2021-04-17 08:30:12 | 28.9400 | 20.9856 | 2.7 ±0.6 | 8.6 | Oil exploitation |
| Australia | a | 2021-06-08 23:23:08 | -21.8321 | 148.0099 | 0.7 ±0.2 | 6.4 | Coal mine |
| | b | 2021-06-08 23:23:08 | -21.8847 | 147.9737 | 1.1 ±0.3 | 6.4 | Coal mine |
| | c | 2021-06-08 23:23:08 | -21.8883 | 147.9949 | 2.9 ±0.8 | 6.4 | Coal mine |
| Uzbekistan | a | 2021-12-20 06:22:11 | 38.7989 | 64.6410 | 6.4 ±1.8 | 6.5 | flaring stack |
| | b | 2021-12-20 06:22:11 | 38.7839 | 64.6289 | 4.0 ±1.1 | 6.5 | O&G facility |
| | c | 2021-12-20 06:22:11 | 38.7347 | 64.6169 | 3.6 ±1.0 | 6.5 | O&G facility |
| Louisiana, US | a | 2021-04-19 16:04:48 | 32.1979 | -93.4882 | 0.5 ±0.4 | 0.1 | O&G facility |
| | b | 2021-04-19 16:04:48 | 32.1960 | -93.4996 | 2.3 ±2.0 | 0.1 | O&G facility |
| | c | 2021-04-19 16:04:48 | 32.1900 | -93.4856 | 0.4 ±0.3 | 0.1 | O&G facility |
| Russia | a | 2021-08-25 04:26:02 | 54.6520 | 86.1506 | 1.3 ±0.6 | 2.8 | Coal facility |
| | b | 2021-08-25 04:26:02 | 54.6313 | 86.1736 | 0.3 ±0.1 | 2.8 | Coal |
| | c | 2021-08-25 04:26:02 | 54.6190 | 86.1747 | 2.4 ±1.1 | 2.8 | Coal |
| | d | 2021-08-25 04:26:02 | 54.6120 | 86.1498 | 1.6 ±0.7 | 2.8 | Coal |
| | e | 2021-08-25 04:26:02 | 54.6122 | 86.1315 | 0.9 ±0.4 | 2.8 | Coal |
| | f | 2021-08-25 04:26:02 | 54.6080 | 86.1496 | 0.5 ±0.2 | 2.8 | Coal |
| | g | 2021-08-25 04:26:02 | 54.5964 | 86.1828 | 0.4 ±0.2 | 2.8 | Coal |
| | h | 2021-08-25 04:26:02 | 54.6168 | 86.1465 | 1.0 ±0.4 | 2.8 | Coal |
| | i | 2021-08-25 04:26:02 | 54.5904 | 86.1812 | 0.2 ±0.1 | 2.8 | Coal |
| | j | 2021-08-25 04:26:02 | 54.5621 | 86.2065 | 0.4 ±0.2 | 2.8 | Coal |

**Table B1.** Observation, location and quantification details corresponding to the GHGSat scenes. One of which is Morocco, Casablanca, Landfill in Figure 7. Windspeeds are 10m windspeeds obtained from GEOS-FP (Molod et al., 2012).

Locations Uzbekistan-b and Uzbekistan-c are located just over the border in Turkmenistan, however the main building of the facility (near Uzbekistan-a) is located in Uzbekistan. Because there is another location in Turkmenistan we have chosen this naming.



| GHGSat | | Observation time [UTC] | Latitude [deg] | Longitude [deg] | Source rate [t h$^{-1}$] | Wind speed $U_{10}$  [m s$^{-1}$] | Sector |
|---|---|---|---|---|---|---|---|
| India | a | 2021-12-24  03:21:48 | 27.4626 | 95.4788 | 1.2 ±0.8 | 0.6 | Oil exploitation |
| | b | 2021-12-24  03:21:48 | 27.3902 | 94.4695 | 2.3 ±1.6 | 0.6 | Oil exploitation |
| | c | 2021-12-24  03:21:48 | 27.3545 | 95.4733 | 2.8 ±2.0 | 0.6 | Oil exploitation |
| | d | 2021-12-24  03:21:48 | 27.3402 | 95.4830 | 3.1 ±2.2 | 0.6 | Oil exploitation |
| | e | 2021-12-24  03:21:48 | 27.3771 | 95.4472 | 3.5 ±2.5 | 0.6 | Oil exploitation |

**Table B1.** Observation, location and quantification details corresponding to the GHGSat scenes. One of which is Morocco, Casablanca, Landfill in Figure 7. Windspeeds are 10m windspeeds obtained from GEOS-FP (Molod et al., 2012).

| PRISMA | | Observation time [UTC] | Latitude [deg] | Longitude [deg] | Source rate [t h$^{-1}$] | Wind speed $U_{10}$  [m s$^{-1}$] | Sector |
|---|---|---|---|---|---|---|---|
| Turkmenistan | a | 2021-02-13  06:51:45 | 40.0106 | 60.9346 | 3.0 ±1.0 | 1.9 | O&G exploitation |
| | b | 2021-02-13  06:51:45 | 40.0496 | 61.0456 | 3.5 ±1.1 | 1.9 | O&G exploitation |
| | c | 2021-02-13  06:51:45 | 40.0240 | 61.0536 | 2.5 ±0.8 | 1.9 | O&G exploitation |
| Shanxi, China | | 2021-12-22  03:18:33 | 35.6083 | 112.5282 | 11.6 ±3.7 | 2.7 | Coal facility |

**Table B2.** Observation, location and quantification details corresponding to the PRISMA scenes. Windspeeds are 10m windspeeds obtained from GEOS-FP (Molod et al., 2012).

The plume mask of plume Turkmenistan-c (Figure 9, Table B2) was curated in order to exclude an artifact which was caused by a nearby road. Retrieval artefacts in high-resolution methane retrievals from hyper-spectral instruments resulting from surface features such as roads is a known issue (Sánchez-García et al., 2022; Gorroño et al., 2022). PRISMA and Sentinel-2 are more prone to such issues than GHGSat-Cx.





| Sentinel-2 | | Observation time [UTC] | Latitude [deg] | Longitude [deg] | Source rate [t h⁻¹] | Wind speed $U_{10}$ [m s⁻¹] | Sector |
|---|---|---|---|---|---|---|---|
| Kazakhstan | a | 2021-05-14 07:15:51 | 49.7205 | 59.0809 | 179.8 ±59.1 | 2.6 | Natural gas pipeline |
| | b | 2021-05-14 07:15:37 | 49.3704 | 59.0660 | 74.6 ±23.0 | 2.2 | Natural gas pipeline |
| Iran | a | 2021-08-24 07:12:03 | 27.5345 | 53.2946 | 10.0 ±3.0 | 2.1 | O&G facility |
| | b | 2021-09-08 07:11:57 | 27.5345 | 53.2946 | 16.1 ±5.3 | 2.7 | O&G facility |
| | c | 2021-10-23 07:12:06 | 27.5345 | 53.2946 | 1.3 ±0.4 | 1.8 | O&G facility |

**Table B3.** Observation, location and quantification details corresponding to the Sentinel-2 scenes. One of which is Kazakhstan, Natural gas pipeline in Figure 8. Windspeeds are 10m windspeeds obtained from GEOS-FP (Molod et al., 2012).





| TROPOMI | | Observation time [UTC] | Latitude [deg] | Longitude [deg] | Source rate [t h⁻¹] | Wind speed GEOS-10m / GEOS-PBL / ERA5-10m [m s⁻¹] | Sector estimate (bottom-up) |
|---|---|---|---|---|---|---|---|
| Casablanca, Morocco | | 2021-06-12 13:48:37 | 33.48 | -7.54 | 14.6 ±8.8 | 6.9 / 2.6 / 2.8 | Landfill |
| Madrid, Spain | | 2021-01-05 13:13:19 | 40.30 | -3.64 | 6.8 ±2.6 | 0.6 / 0.6 / 1.0 | Landfill |
| Libya | | 2021-07-26 11:40:42 | 28.88 | 20.93 | 87.3 ±28.7 | 4.4 / 4.6 / 3.4 | Oil |
| Australia | | 2021-09-25 03:56:01 | -21.91 | 148.06 | 17.5 ±6.1 | 3.0 / 2.0 / 2.5 | Coal |
| Uzbekistan | | 2021-08-10 08:39:36 | 38.76 | 64.59 | 15.2 ±5.6 | 4.2 / 3.6 / 4.9 | Gas |
| Louisiana, US | | 2021-09-23 20:02:42 | 32.13 | -93.72 | 31.1 ±12.1 | 1.1 / 1.9 / 1.6 | Gas |
| Russia | | 2021-06-08 06:43:36 | 54.61 | 86.10 | 40.4 ±16.9 | 1.1 / 1.8 / 1.0 | Coal |
| India | | 2021-01-04 06:42:43 | 27.38 | 95.69 | 21.1 ±9.0 | 1.0 / 0.8 / 0.5 | Gas |
| Turkmenistan | | 2021-12-05 08:47:19 | 40.10 | 60.99 | 11.1 ±3.9 | 2.4 / 3.2 / 3.3 | Oil |
| Shanxi, China | | 2021-01-23 05:47:47 | 35.63 | 112.52 | 47.7 ±15.3 | 3.5 / 4.6 / 3.6 | Coal |
| Kazakhstan | a | 2021-05-14 07:53:19 | 49.85 | 59.12 | 35.2 ±13.2 | 2.2 / 2.5 / 1.4 | Gas |
| Kazakhstan | b | 2021-05-14 07:53:14 | 49.46 | 59.08 | 28.1 ±11.2 | 2.3 / 3.0 / 1.5 | Gas |
| Iran | | 2021-11-04 10:05:30 | 27.48 | 53.41 | 45.4 ±13.7 | 2.1 / 1.8 / 2.0 | Oil |

**Table B4.** Observation, location and quantification details of the TROPOMI scenes corresponding to the high resolution observations in Figures 7, 8 and 9. Windspeeds presented in this table are the GEOS-10m, GEOS-PBL and ERA5-10m windproducts (Molod et al., 2012; Hersbach et al., 2020) which are used to compute three $U_{\mathrm{eff}}$ values, which are then averaged (Section 2.3.1).





**570   Appendix C:  Features used as input for the SVC**





| Feature name | Category | FIR | Value range | Description |
|---|---|---|---|---|
| CNN_score | plume-like morphology | 1 | [0.5, 1] | prediction score assigned by the CNN |
| valid_pixels | quality of the scene | 7 | [0.2, 1] | fraction of valid pixels in the scene, N/(32x32) |
| count_mask_high | plume mask | | [1, > | number of pixels in the high confidence plume mask |
| sum_mask_high | plume mask | | [1, > | sum of enhancement of the pixels in the high confidence plume mask |
| stdev | xch4 statistical distribution | | [0, > | standard deviation of the xch4 value of all pixels in the scene |
| skew | xch4 statistical distribution | | -2, > | skewness of the xch4 value of all pixels in the scene |
| kurtosis | xch4 statistical distribution | | [1, > | kurtosis of the xch4 value of all pixels in the scene |
| IME | magnitude of plume | | [0, > | [kg] Integrated Mass Enhancement (section 2.3.1) |
| L | magnitude of plume | | [0, > | [m] plume length (section 2.3.1) |
| U_10 | wind | | [0, ~8] | [ms$^{-1}$] 10 meter windspeed, obtained from ERA5, present in Level 2 dataproduct |
| cba_sum | clouds | | [0, > | sum product of a 3x3 kernel multiplying enhancements of the high conf plume with cloud fraction |
| cba_count | clouds | | [0, > | number of pixels in the high confidence plume mask close to cloudy pixels based on 3x3 kernel |
| angle_PCA_mean_wind _plume_mask_weighted | wind | 8 | [0, 90] | [deg] angle between principle axis of the plume mask and mean windvector (section 2.3) |
| exp_var_ratio _pca_weighted | plume elongation | | [0, > | ratio between the variance along the primary and secondary axis |
| albedo_rvalue_bg | correlation xch4 & supporting data | | [-1, 1] | Pearson r-value between xch4 and albedo for the full scene |
| albedo_rvalue_dil_1 | correlation xch4 & supporting data | 3 | [-1, 1] | Pearson r-value between xch4 and albedo for pixels within 1 dilation around the low confidence mask |

**Table C1.** Overview of the features used by the SVC as input. Each scene is represented as a feature vector with shape [41x1]. This table provides the name of the feature, the category the feature is aimed to provide information about, the ranking of the top-10 features in the feature importance analysis (FIR = Feature Importance Ranking), the possible range of values the feature can attain and a description of the feature.





| Feature name | Category | FIR | Value range | Description |
|---|---|---|---|---|
| aero_rvalue_bg | correlation xch4 & supporting data | | [-1, 1] | Pearson r-value between xch4 and AOT for the full scene |
| aero_rvalue_dil_1 | correlation xch4 & supporting data | | [-1, 1] | Pearson r-value between xch4 and AOT for pixels within 1 dilation around the low confidence mask |
| surf_pres_rvalue_bg | correlation xch4 & supporting data | | [-1, 1] | Pearson r-value between xch4 and surface pressure for the full scene |
| surf_pres_rvalue_dil_1 | correlation xch4 & supporting data | | [-1, 1] | Pearson r-value between xch4 and surface pressure for pixels within 1 dilation around the low confidence mask |
| chi2_rvalue_bg | correlation xch4 & supporting data | 5 | [-1, 1] | Pearson r-value between xch4 and chi2 for the full scene |
| chi2_rvalue_dil_1 | correlation xch4 & supporting data | 0 | [-1, 1] | Pearson r-value between xch4 and chi2 for pixels within 1 dilation around the low confidence mask |
| cloud_angle_high | clouds | | [0, 90] | angle of the principle axis of the high confidence plume mask with the principle axis of a cloud |
| cloud_angle_low | clouds | | [0, 90] | angle of the principle axis of the low confidence plume mask with the principle axis of a cloud |
| coast_angle | surface conditions | | [0, 90] | angle of the principle axis of the high confidence plume mask with a coast |
| avg_chi2_mask_high | supporting data | 4 | [0, > | average chi2 value of the pixels within the high confidence plume mask |
| avg_chi2_mask_low | supporting data | | [0, > | average chi2 value of the pixels within the low confidence plume mask |
| avg_albedo_mask_high | supporting data | | [0, 1] | average albedo value of the pixels within the high confidence plume mask |
| avg_albedo_mask_low | supporting data | | [0, 1] | average albedo value of the pixels within the low confidence plume mask |
| avg_aot_mask_high | supporting data | | [0, 1] | average AOT value of the pixels within the high confidence plume mask |
| avg_aot_mask_low | supporting data | | [0, 1] | average AOT value of the pixels within the low confidence plume mask |

**Table C1.** Overview of the features used by the SVC as input. Each scene is represented as a feature vector with shape [41x1]. This table provides the name of the feature, the category the feature is aimed to provide information about, the ranking of the top-10 features in the





| Feature name | Category | FIR | Value range | Description |
|---|---|---|---|---|
| avg_QA_mask_high | supporting data | 2 | [0.4, 1] | average QA value of the pixels within the high confidence plume mask |
| avg_QA_mask_low | supporting data | 9 | [0.4, 1] | average QA value of the pixels within the low confidence plume mask |
| std_bg_xch4_high | background homogeneity | | [0, > | standard deviation of the xch4 outside of the high confidence plume mask, similar to pixel precision Varon2021 |
| std_bg_xch4_low | background homogeneity | | [0, > | standard deviation of the xch4 outside of the low confidence plume mask, similar to pixel precision Varon2021 |
| avg_enh_plumemask_high_above_bg | magnitude of plume | | [0, > | average enhancement above the background of the pixels within the high confidence plumemask |
| avg_enh_plumemask_low_above_bg | magnitude of plume | | [0, > | average enhancement above the background of the pixels within the low confidence plumemask |
| max_enh_plumemask_high_above_bg | magnitude of plume | 6 | [0, > | maximum enhancement above the background of the pixels within the high confidence plumemask |
| frac_land_pixels_high | surface conditions | | [0, 1] | fraction of pixels with surface classification 'land' in the high confidence plume mask |
| frac_landwater_pixels_high | surface conditions | | [0, 1] | fraction of pixels with surface classification 'land+water' in the high confidence plume mask |
| frac_coast_pixels_high | surface conditions | | [0, 1] | fraction of pixels with surface classification 'coast' in the high confidence plume mask |

**Table C1.** Overview of the features used by the SVC as input. Each scene is represented as a feature vector with shape [41x1]. This table provides the name of the feature, the category the feature is aimed to provide information about, the ranking of the top-10 features in the feature importance analysis (FIR = Feature Importance Ranking), the possible range of values the feature can attain and a description of the feature.



*Author contributions.* BJS, JDM, and IA designed the study. BJS performed the TROPOMI analysis with contributions from PB, GM, SP, and JDM. BJS, JDM, and IA wrote the paper with contributions from all authors. AWB and SH provided the WRF simulations used to calibrate the TROPOMI IME method. AL and TB provided the TROPOMI methane data and associated support. DJV, JMcK, DJ, and MG provided the GHGSat data and supported the GHGSat analysis. II, JG, and LG provided the Sentinel-2 data/analysis. DHC provided the
PRISMA data/analysis.

*Competing interests.* The authors declare that they have no competing interests.

*Acknowledgements.* We thank the team that realized the TROPOMI instrument and its data products, consisting of the partnership between Airbus Defence and Space Netherlands, KNMI, SRON, and TNO, commissioned by NSO and ESA. Sentinel-5 Precursor and Sentinel-2 are part of the EU Copernicus program, and Copernicus (modified) Sentinel-5P data (2018-2021) have been used as well as (modified)
Sentinel-2 data (2021). The TROPOMI data processing was carried out on the Dutch national e-infrastructure with the support of SURF Cooperative. Financial support is acknowledged from the NSO TROPOMI national programme for AL, the GALES project (15597) by the Dutch Technology Foundation STW-NWO for SP, and ESA through EDAP for GM.



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
