# Peer review of "Automated detection and monitoring of methane super-emitters using satellite data"

_Atmospheric Chemistry and Physics, 2022_

## Author Comment (AC1)

We thank both reviewers for their constructive comments and feedback on how to improve both the quality and clarity of the manuscript. We implemented the comments below and also made minor changes and improvements elsewhere to improve the flow of the manuscript.

Grey are the reviews, replies to the reviews are in black, *blocks of text in the manuscript are italic within " "* , **additions to this are italic and bold.**

**Anonymous Referee #1, 21 Feb 2023**
**https://doi.org/10.5194/acp-2022-862-RC1**

This paper describes a two-step machine learning approach that uses a Convolutional Neural Network (CNN) to detect plume-like structures in TROPOMI methane data and then applies a Support Vector Classifier (SVC) to distinguish emission plumes from retrieval artefacts. The CNN is trained using hand-selected scenes from 2018-2020 and then applied to 2021 observations. This is an important topic because TROPOMI collects millions of measurements over the globe each day, and future missions will collect even more. Automated approaches are therefore needed to process these data and reliably identify emission plumes. In general, this manuscript does represent a substantial contribution. However, the methods section (section 2.2, 2.3) needs a substantial revision to make it more understandable to the average reader of Atmospheric Chemistry and Physics, who may not be familiar with these machine learning techniques.

We thank the reviewer for this positive overall review. The feedback on the methods section is implemented as discussed in more detail below. We have added a flowchart (new Figure 1), which illustrates the different steps in our method, what TROPOMI data are used in every step, what the format of the data is (32x32 scene vs. 1x41 feature vector), when a model is trained, where a priorly trained model is used and which steps require manual input. This flowchart is referenced at various locations in the manuscript.

The description of the choice and configuration of the CNN is very short. Four reasons are cited to justify the choice of this particular machine-learning method (L131-136), but readers not familiar with these methods might not know that CNNs are commonly chosen for image recognition and pattern recognition. We learn that "the same convolutional kernel scans the entire image", but never told what this kernel is or where it comes from or what a "pooling layer" (figure 2) is or does.

We have added an introductory section on CNNs at the start of the methodology section and also include a reference to an overview paper about deep learning methodologies applied to remote sensing images to give the context that CNNs are commonly used for image recognition.

*"Convolutional Neural Networks (CNN) are a type of machine learning model commonly applied in image recognition and object detection problems (Cheng et al., 2020). A CNN consists of multiple layers, where information moves from an input image, through the layers of the CNN at an increasingly abstract coarse resolution, to the output; the classification of the image. To condense the information of the image to coarser resolution, the CNN uses "convolutional blocks" that consist of two or more convolutional layers, followed by a (max-)pooling layer. A convolutional layer produces "feature maps" that indicate where certain features (e.g. curves, edges or more abstract features) are detected within the image. These feature maps are obtained by convoluting the input image with a convolutional kernel, a small matrix with weights that are optimised during training to best detect features relevant to the particular classification problem. The resulting feature maps (one for each kernel applied) are then the input for the next layer (LeCun et al., 2010; Cheng et al., 2020). A max-pooling layer scans the previous layer with a 2x2 kernel and returns the maximum value, thereby creating a feature map at half the resolution, focused on dominant features (LeCun et al., 2010). After the last convolutional block, the resulting feature maps are flattened and interpreted by one or more fully-connected "dense" layers, consisting of neurons, between which the connections have trainable weights. This part of the network aggregates the information into a single output value. During training, the trainable weights, in the convolutional kernels and dense layers, are optimized to best perform the classification task based on the training dataset (LeCun et al., 2010; Cheng et al., 2020). The trained CNN can then be used to classify new images, in this case as 'plume' versus 'no-plume'."*

We now elaborate on what a convolutional kernel and a pooling layer are in the general introduction on CNNs (see above) and improved the justification for the choice for the CNN together with additional CNN references:

*"The main advantages of the CNN compared to regular neural networks or other machine learning models are that: (1) the CNN is capable of **better** retaining spatial information which is lost in fully-connected **networks** or machine learning models like decision trees or support vector machines (Selvaraju et al., 2020); **(2) the training of a CNN can be done with image level labels (plume or no plume), there is no need to indicate where within the image the feature of interest is located as the CNN learns to localize these features during training; (3)** the same convolutional **kernels are convoluted with** the entire image, which is **more** computationally efficient **compared to fully-connected networks (LeCun et al., 2010; Cheng et al., 2020); (4)** the model is rotational and translational invariant when properly trained (LeCun et al., 2010)."*

> The description of the CNN training process is even more obscure and confusing. We are told that "For the training process, the class weight parameter is set to the ratio between the number of plumes (828 positives) and negatives (2242), …", but we are never told what the class weight parameter is or how sensitive the solution might be to this setting.

We have added an explanation on class weights with a detailed reference that specifically covers the issue of class imbalance:

*"**Our** training **dataset** mostly **contains** clear positives **(828)** and clear negatives **(2242)** to effectively learn distinguishing features. **Our dataset has many more negative than positive scenes. When training our CNN however, we want both categories of training samples (classes) to have equal impact to obtain optimal performance (Johnson and Khoshgoftaar, 2019). This balancing can be achieved by applying class weights during training (Johnson and Khoshgoftaar, 2019), giving positive scenes more weight. We set the class weight parameter to the inverse of the ratio between the number of positives and negatives.*"*

The paragraph that follows (L148-163) makes the training process look more like black magic, where the user utters a few magic words (Keras, ReLU, ADAM, softmax) and wondrous things happen. All of these terms are used without reference to refereed scientific papers.

Instead, the reader is sent to a web page (Chollet et al., 2015) with a sales pitch and code and then a github site (O\'Malley et la.. 2019). What part of these code distributions are used here?  All of them? The only real reference in this paragraph is Li et al. (2018), which describes one of two approaches used for optimizing hyperparameters. Neural network training is a major of this paper. Additional insight into these methods is essential to gain the acceptance and understanding of this Earth Science audience. At a minimum, we need to understand the specific inputs and outputs of these methods and how the results are validated against standards.  A few additional  figures illustrating these topics would be great.

We now added a flowchart (new Figure 1) to provide an overview of where the CNN training process fits within the entire ML approach, and provide additional details on inputs, output and data formats in this step. We have added additional references to academic literature where possible in this paragraph (e.g. for ADAM). For Keras and KerasTuner we have also maintained the (web-based) references requested by their authors. We wrote a more elaborate discussion on the function of different components of the CNN and better explained the different terms. We fixed an error where we wrote we used softmax as activation function in the output neuron, this was changed to sigmoid.

[Figure]

*"Figure 1: Flowchart showing the employed methodology framework. It consists of three phases, where each next phase uses the output (the trained model) of the previous phase as indicated with matching colors (orange and green). We use the TROPOMI XCH₄ Level 2 scientific data product version 18_17. Pre-processing is equivalent for each of the three phases, and consists of filtering, de-striping of the XCH₄ channel and splitting up the data into 32x32 scenes. The output of the pre-processing is a dataset of [N, M, 32, 32] where N is the number of scenes, M is the number of channels (fields of data used later on, for example the methane concentrations) and 32x32 gives the (pixel) dimensions of the scene. The CNN exclusively uses the XCH₄ channel, both during training and when the trained CNN is used for classification. In the Feature Engineering step of the second and third phase, a feature vector of shape [1, 41] is computed which corresponds to a single scene [1, M, 32, 32]. The SVC exclusively uses the feature vectors, both during training and when the trained SVC is used for classification. Manual verification steps are shown in purple. In the application phase, there is one manual step, which is the verification of detected plumes to make sure the output of the pipeline is correct."*

*"We use the training dataset of 19,648 (augmented) scenes (Table A1) to train the CNN **(Figure 1, 'CNN Training')**. The CNN was designed and trained **using the machine learning framework** Keras (Chollet et al. (2015), **Chollet (2021)**), at first **using** the default values for the hyperparameters. The model is trained for a maximum of 100 epochs **(iterations of the training process)**. **During training we optimize** the validation loss, **which measures the error made on the subset of the training data not used in that epoch. We use** binary cross-entropy as the loss function and ADAM **(an improved version of stochastic gradient descend algorithm, Kingma and Ba (2014))** as the optimizer. We use a 0.4 dropout layer **(randomly disabling 40% of the neurons)** in the first fully-connected layer during training to **prevent overfitting and** make the model more robust **(Srivastava et al., 2014). The activation function modifies the output before it is passed to the next layer; we apply** the ReLU (rectified linear unit**, outputs zero when the input is negative and otherwise outputs the input value)** activation function in all layers except for the final layer where we apply **sigmoid (LATEX MATH), which normalizes the output.** To force the model to focus **more** on*

*plume-like signatures **during training**, the loss weight of plume scenes is set to double that of negatives scenes. Training is halted after the validation loss does not improve for several epochs, the best model weights **found up to that point** are **used**. After training, the model performance is inferred by classifying the labeled **test dataset**.*

*After training this initial "default" model version, the hyperparameters were further optimized using **the** Keras Tuner **optimization framework** (O'Malley et al., 2019) and Hyperband (Li et al., 2018). **With these methods, we perform a grid-search to find the best hyperparameters for our particular problem.** The optimal hyperparameters depend on the size of the training dataset, architecture of the CNN, number of classes and problem type. The search space for the optimization was defined using insights from the initial training, theoretical foundation, and design constraints. We inspected the hyperparameters of the top 10 performing **set-ups** and selected the optimal hyperparameters by combining **the results of** this optimization with expert judgement on this particular problem. Figure **3** shows a schematic overview of the CNN with optimized hyperparameters."*

The discussion of Feature Engineering (Section 2.3, L193-199) and list of features in Table C1 is more helpful, but still unnecessarily confusing. I was surprised that the feature vector included the CNN score (0, 1) as feature, that is apparently no less or more important than any other. We are told that the algorithm operates on 32x32 pixel scenes using a 41 x 1 feature vector. However, then we learn (L205) that "In our binary classification problem, the CAM visualizes which regions of the deepest feature maps(the 8x8, deepest max-pooling layer in Figure 2) lead to an activation of the plume class." Figure 2 shows only two pooling layers and does not mention where the 8x8, deepest max-pooling layer. We are then told (L207-208) that "This spatial activation is calculated using the gradients between all internal 64 feature maps and the fully-connected layer." Where did we learn about internal 64 feature maps? At this point, I was totally lost.

We have added a statement on the fact that the output CNN score is also valuable to the SVC:

*"Scenes with prediction scores >0.5 are classified as plumes. **Although we use this output for binary classification, the value holds additional information regarding the confidence of the CNN (i.e. 0.6 vs. 0.98), which we use for the second model.**"*

We now state more clearly which steps use a 32x32 pixels scene and which steps use a 1x41 feature vector, both in the flowchart and at relevant locations throughout the text. Feature maps are now also explained in the CNN introduction above, the difference between a feature map and feature vector, which are unrelated, is now more clearly reflected in the text and in the flowchart.

We changed the figure with the CNN architecture (now figure 3) to more clearly show which layers are used for the CAM (Max-Pooling 2 and Dense 1). We changed the naming of the layers and referred to those layers more clearly in the text.

*"Fundamental to many of those features is "masking" the plume in order to isolate the plume pixels from the background. For this purpose**,** we use **information about** which part of the scene has triggered the CNN detection. **For this we use the** Class Activation Map (CAM) to*

*visualize the localized activations of a CNN corresponding to a certain class on which it was trained (Zhou et al., 2015). We apply Grad-CAM (Selvaraju et al., 2020), which allows the computation of the CAM for our CNN that includes fully-connected layers. In our binary classification problem, the CAM visualizes which regions of the deepest **(coarsest)** feature maps **(Max-Pooling 2 [8x8]** in Figure 3) **contribute strongest** to an activation of the plume class **(output > 0.5) for a given input image**. This spatial activation is calculated using the gradients **(Selvaraju et al., 2020)** between **the** 64 feature maps **(each of 8x8 resolution, resulting in a 64x8x8 array, Figure 3) of the deepest max-pooling layer (Max-Pooling 2)** and the **first** fully-connected, **or dense** layer **(Dense 1 in Figure 3)**. In order to obtain a CAM of sufficient resolution…"*

Smaller issues, concerns and editorial suggestions:
L66: "As such the hyperspectral …" à "As such, the hyperspectral …"

This was changed in the text following the reviewer's suggestion.
*"As such**,** the hyperspectral PRISMA instrument…"*

L78: "atmospheric conditions and identify plume signatures" à "atmospheric conditions to identify plume signatures …"?

This was changed in the text following the reviewer's suggestion.
*"…include monitoring anomalous atmospheric conditions **to** identify plume signatures in large datasets…"*

L85: "we target three high-resolution satellite instruments …"  The word "target" is ambiguous here, because it sometime means that you point a satellite instrument (i.e., GHGSat) at a target.  From the context, I believe you mean "we use data from three high-resolution satellite instruments …"  Is that correct?

We changed the text to remove the ambiguity between targeted observations and targeted analysis:

*"Based on the 2021 detections, we **use observations of** three high-resolution satellite instruments…"*

L90: "We use two 90 machine learning models in sequence to detect plumes in the TROPOMI methane data. First we apply a Convolutional Neural Network to detect plume-like structures in TROPOMI methane atmospheric mixing ratio data, then we use additional atmospheric parameters and supporting data to further distinguish between genuine methane plumes and retrieval artefacts. We then use (targeted) high-resolution methane observations to pinpoint the responsible sources."
These two sentences are largely redundant with the last few sentences of the paragraph above. It would be better to replace them with a brief overview of the next few subsections.

This was done in order to make sure the Methodology section was self-contained. With the addition of the flowchart at the start of the Methods section we modified this paragraph to incorporate a brief outline of the flowchart and Methods section:

*"Figure 1 illustrates the full machine-learning pipeline and training process. Section 2.1 describes the pre-processing step to generate scenes used by the CNN and the feature engineering algorithms. Section 2.2 describes the training process of the CNN (Figure 1 'CNN Training'). Section 2.3 describes the feature engineering algorithms, which are used to generate feature vectors for each TROPOMI scene. The SVC uses those feature vectors during its training process (Figure 1 'SVC Training'), covered in Section 2.4. Then, we apply the full, trained, machine learning pipeline to 2021 TROPOMI observations the models have not been trained on (Figure 1 'CNN+SVC Application'). Based on the resulting TROPOMI detections, we perform further analysis (Section 2.5) and use (targeted) high-resolution methane observations (Section 2.6) to pinpoint the responsible sources for 12 of those detections."*

Sections 2.2 – 2.4 – see comments above.
L314: "allows multiple close by plumes" à "allows multiple nearby plumes"
This was changed in the text following the reviewer's suggestion.
*"…are cut in half and allows multiple **nearby** plumes to be detected in adjacent…"*

Also, somewhere in this paragraph, it would be good to specify the minimum number of enhanced pixels needed to define a "plume".
We added a statement on the minimum number of enhanced pixels in a plume mask in the section about the plume mask.
*"…from the plume mask. **A plume mask can consist of any number of pixels depending on the scene. The minimum is 1 pixel, but this is rare, the average is around 20 pixels.** Several statistics of the…"*

L345, L358, L371. It appears that the same, 10m wind field is used in the analysis of GHGSat, PRISMA, and sentinel-2, but two different notations are used. For GHGSat (L345), it is called with U10 winds from GEOS-FP, while for the other two satellites, it is called "GEOS-FP 10m wind data". It would be good to use consistent nomenclature.
This was changed in the text following the reviewer's suggestion, we changed "*U10 winds*" to "*10m wind data*" in the GHGSat section. In the IME section we use $U_{10}$ in the equations, we explicitly defined $U_{10}$ and $U_{PBL}$ in that same section for clarity. Everywhere else in the text we now write 10m wind data.

L351: "hyperspectral 30x30 km2 images at a spatial resolution of 30x30 m …" If the pixels and images are nearly square, it would be better to describe their dimensions as 30 km x 30 km and 30 m x 30 m, respectively.
This was changed in the text following the reviewer's suggestion. We also changed this notation elsewhere in the text, for example when stating GHGSat's resolution.

L352: "The minimum revisit time can be up to 7 days with ±20\% across-track pointing."

> Do you mean "up to" or "as short as" 7 days? For example, can PRISMA sometimes have repeat cycles as short as one day or are they almost always longer than 7 days?

Thank you for pointing this out, this was indeed phrased in an unclear way. The revisit time is at least 7 days and even longer when only considering nadir viewing geometry. We added a reference to the original paper about the instrument which mentions the revisit time for this viewing geometry:

*"The revisit time can be **as short as** 7 days **using the instrument's** +/- 20% across-track pointing **(Cogliati et al., 2021)**."*

> L356: "location of interest on a future moment in time." à "location of interest in the future."

This was changed in the text following the reviewer's suggestion.

*"…used to target a location of interest **in the** future. We perform…"*

> L361: "capable of the detection of methane" à "capable of detecting methane"

This was changed in the text following the reviewer's suggestion.

*"…to be capable of **detecting** methane super-emitter plumes…"*

> L362: "with a pixel resolution of 20 m" à is this "with a pixel resolution of 20 m x 20 m"?

This was changed in the text following the reviewer's suggestion, also elsewhere in the text.

*"…Instrument, with a pixel resolution of **20 m x** 20 m for the…"*

> L369: "that similarly to Varon et al. (2021)" à "", that, like Varon (2021) uses ..."

This was changed in the text following the reviewer's suggestion:

*"We apply the methane retrieval and IME quantification approach from Gorroño et al. (2022**), that, like** Varon et al. (2021) uses a "reference day" without a…"*

> L375: "identifies 26,444 scenes (3.3 \%) as containing" à "identifies 26,444 scenes (3.3 \%) that contain"

This was changed in the text following the reviewer's suggestion.

*"…scenes, the CNN identifies 26,444 scenes (3.3 %) **that contain** plume-like XCH4 morphological…"*

**Anonymous Referee #2, 06 Mar 2023**
**https://doi.org/10.5194/acp-2022-862-RC2**

General comments
This study by Schuit et al. proposed a machine-learning-based approach to automatically detect super-emitters of methane from high-resolution TROPOMI retrievals. This is an important work because it provides opportunities to make the most of the huge amount of measurements obtained by TROPOMI or other instrument in the future. However, I found the methodology section hard to follow, as it involves different models (e.g., CNN, SVC, IME, the Weather Research and Forecasting model coupled with a Chemistry module, version 4.1.5), different datasets (e.g., TROPOMI, emission inventory, GEOS-FP, GHGSat, PRISMA and Sentinel-2), and the relationship between these models and datasets are complicated.

A flow chart of the full methodology framework is needed, which describes each step of the approach in detail, including the purpose, the input and output data, as well as the involved model. Such figure would help the readers to better understand the whole procedures of the complicated approach. For example, it could explain the relationship between the CNN and the SVC model, and describes how the data is transferred between these models.
The method section is suggest to be reorganized according to the method flow chart.

We thank the reviewer for the positive review. We recognize the complexity of the different models, datasets and instruments and different steps of the Methodology. We have added a flowchart (new Figure 1) as well as a more comprehensive introduction and overview of the methodology at the start of the Data and Methods section to further clarify the structure and contents of the paper, and how different parts relate to one another. The flowchart clearly splits the methodology section into three parts (training of the CNN, training of the SVC and operational use) and shows the relations between the three parts (i.e. how the trained models are applied to later steps). Further specific comments are addressed below.

[Figure]

*"Figure 1: Flowchart showing the employed methodology framework. It consists of three phases, where each next phase uses the output (the trained model) of the previous phase as indicated with matching colors (orange and green). We use the TROPOMI XCH₄ Level 2 scientific data product version 18_17. Pre-processing is equivalent for each of the three phases, and consists of filtering, de-striping of the XCH₄ channel and splitting up the data into 32x32 scenes. The output of the pre-processing is a dataset of [N, M, 32, 32] where N is the number of scenes, M is the number of channels (fields of data used later on, for example the methane concentrations) and 32x32 gives the (pixel) dimensions of the scene. The CNN exclusively uses the XCH₄ channel, both during training and when the trained CNN is used for classification. In the Feature Engineering step of the second and third phase, a feature vector of shape [1, 41] is computed which corresponds to a single scene [1, M, 32, 32]. The SVC exclusively uses the feature vectors, both during training and when the trained SVC is used for classification. Manual verification steps are shown in purple. In the application phase, there is one manual step, which is the verification of detected plumes to make sure the output of the pipeline is correct."*

The new start of the Data and Methods section now includes references to the specific sections, linking it to the overview in the flowchart:

*"Figure 1 illustrates the full machine-learning pipeline and training process. Section 2.1 describes the pre-processing step to generate scenes used by the CNN and the feature engineering algorithms. Section 2.2 describes the training process of the CNN (Figure 1 'CNN Training'). Section 2.3 describes the feature engineering algorithms, which are used to generate feature vectors for each TROPOMI scene. The SVC uses those feature vectors during its training process (Figure 1 'SVC Training'), covered in Section 2.4. Then, we apply the full, trained, machine learning pipeline to 2021 TROPOMI observations the models have not been trained on (Figure 1 'CNN+SVC Application'). Based on the resulting TROPOMI detections, we perform further analysis (Section 2.5) and use (targeted) high-resolution methane observations (Section 2.6) to pinpoint the responsible sources for 12 of those detections."*

Another important information that should be reflected in the flow chart is, whether the specific step is automatic or manual. I've noticed that some processes in this method need manual inspection, and such kind of information are suggested to be summarized.

We have explicitly listed the three manual labeling/verification steps in the flowchart, and color-coded those steps purple. We also clarified this where necessary elsewhere in the text, e.g. in the Conclusions:

*"We have trained a Convolutional Neural Network with a relatively small set of **manually identified** plumes in pre-2021 TROPOMI methane data to detect plume-like morphological structures (kappa score of 0.97 and recall of 0.98 on the test set)."*

The evaluation of the estimated emission source rates are conducted through comparison with GHGSat, PRISMA, and Sentinel-2 detections. Such comparison was conducted for several locations as examples. I suggest the authors to clarify the representativeness of these locations. Meanwhile, comparison of all the available source rates estimated from different methods are encouraged, including the correlation coefficient, mean bias, etc.

We now mention how these 12 selected locations show the range of typical source types and intermittencies we have observed:

*"Supplemental Tables B1-B3 provide details on these high resolution observations and associated (not necessarily on the same day) TROPOMI scenes (Table B4). **These 12 selected locations show the range of typical anthropogenic source types and intermittencies we have observed with both TROPOMI and high-resolution instruments.**"*

We do not aim to evaluate the TROPOMI emission estimates with those from the high-resolution instruments. We use TROPOMI detections to guide follow-up targeted observations or analysis with high resolution satellites to identify the main source(s) responsible for these super emissions. The high spatial resolution instruments are particularly sensitive to point source emissions at a scale of ~25 m, while TROPOMI integrates all (diffuse) emissions over its km-scale footprints. We now clarify this aspect in the text.

*"**Because of the different spatial footprints, sensitivities, and detections dates, these emission rates cannot be directly compared to the TROPOMI emission estimates [Maasakkers et al., 2022].**"*

For the nearly concurrent observations of isolated point sources in Kazakhstan, we do discuss the large difference between TROPOMI and high-resolution estimates. We now also more prominently mention the inter-comparison between the high-resolution instruments in a controlled release experiment by Sherwin et al. (2022) in Section 2.6:

*"During a controlled release experiment **comparing methane-observing capabilities of different high-resolution instruments** by Sherwin et al. ( 2022), a plume of ~200 kg/h was successfully detected."*

Other specific comments:
The only third-level heading under Sect. 2.3 is Sect. 2.3.1. Besides, does "source rate quantification" belong to "feature engineering"? I think that the source rate quantification step should be after the SVC model when the artefact is excluded.
We use some of the (intermediate) outputs of the IME method, such as the length of the plume mask, as features for the SVC. We do agree that the detailed quantification (including uncertainty ensemble) better fits later in the paper, therefore we moved Section 2.3.1 to the plume characterization section and briefly refer to that section in the description of the SVC model:
*"…Elongation ratio of the plume, **several intermediate outputs of** the source **rate estimate (Section 2.5.1)** and several statistical properties…"*

And we mention that intermediate output of the IME is used for the SVC in the Source rate quantification section:
*"…method (Frankenberg et al., 2016; Varon et al., 2018). **Some intermediate outputs of the IME method (such as the plume length) are used as features in the feature vector for the SVC (Section 2.4). We perform a full source rate quantification, including uncertainty estimates, for the plumes that pass the machine learning pipeline and are manually verified.** The IME method relates the emission…"*

Line 322: How did the authors define the dominant source type in a specific grid if it contained mixed emission sources?
We further clarified the statement:
*"We identify the dominant source type **as the source type with the largest annual flux** in a 0.7°x0.7° square centered around the estimated source location."*

Line 332: What is the inversion-based method? Please provide a brief description of this method and its comparison with the new method developed in this study.
We now better explain the method employed by Maasakkers et al. 2022a.
*"The estimated source location of the plume is 2.2 km away from the source and our automated quantification estimate is 121 ± 46 t h$^{-1}$. **Our estimate is** in good agreement with the quantification by Maasakkers et al. (2022a) of 101 (49-127) t h$^{-1}$ **who scale a plume simulated with the WRF atmospheric transport model to match the enhancements seen in TROPOMI using a Bayesian inversion.**"*

"Is it able to capture the day-to-day variation of emissions for a specific source detected?"
We now mention the potential of our method to capture temporal (up to day-to-day) variation in the conclusions:
> *"Over the next few years, the number of global, regional, and point-source mapping instruments capable of retrieving methane plumes will vastly increase; with i.a. Sentinel-5, CO2M, MethaneSAT and Carbon Mapper (Jacob et al., 2022). Our monitoring system can incorporate these fast-growing data volumes and can already be used to automatically detect plumes in the operational TROPOMI data, **track temporal variability in super-emitter plumes**, and "tip-and-cue" high-resolution*

*satellite instruments to find the associated super-emitting facilities. This identification and monitoring of super-emitters with large mitigation potential is paramount to reach the goals of the Global Methane Pledge."*